Letter

# Discovery of iridoid cyclase completes the iridoid pathway in asterids

Maite Colinas [1] ✉, Chloée Tymen[1], Joshua C. Wood[2], Anja David[1], Jens Wurlitzer[1], Clara Morweiser [1], Klaus Gase [1], Ryan M. Alam [1], Gabriel R. Titchiner[1], John P. Hamilton [2,3], Sarah Heinicke[1], Ron P. Dirks[4], Adriana A. Lopes[5], Lorenzo Caputi[1], C. Robin Buell [2,6,7] ✉ & Sarah E. O'Connor [1] ✉

Iridoids are specialized monoterpenes ancestral to asterid flowering plants[1,2] that play key roles in defence and are also essential precursors for pharmacologically important alkaloids[3,4]. The biosynthesis of all iridoids involves the cyclization of the reactive biosynthetic intermediate 8-oxocitronellyl enol. Here, using a variety of approaches including single-nuclei sequencing, we report the discovery of iridoid cyclases from a phylogenetically broad sample of asterid species that synthesize iridoids. We show that these enzymes catalyse formation of 7S-cis-trans and 7R-cis-cis nepetalactol, the two major iridoid stereoisomers found in plants. Our work uncovers a key missing step in the otherwise well-characterized early iridoid biosynthesis pathway in asterids. This discovery unlocks the possibility to generate previously inaccessible iridoid stereoisomers, which will enable metabolic engineering for the sustainable production of valuable iridoid and iridoid-derived compounds.

Iridoids are bicyclic monoterpenes widespread among asterid plants[1,2]. Iridoids play important roles in plant defence; volatile iridoids are used by plants to repel or attract insects, whereas glycosylated forms serve as feeding deterrents[3–7]. From a pharmacological perspective, iridoids possess promising anti-inflammatory activity[8] and, moreover, serve as precursors for medicinally important monoterpenoid indole and ipecac alkaloids, which include anti-cancer (camptothecin and vin-blastine), anti-malarial (quinine), putative anti-addiction (ibogaine) and emetic (emetine) agents (recently reviewed by ref. 9).

Nepetalactol, which is the simplest iridoid and the common intermediate for all ~1,000 known iridoids, is biosynthesized from geranyl-diphosphate (GPP). GPP is subjected to dephosphorylation by geraniol synthase (GES), hydroxylation by geraniol 8-hydroxylase (G8H) and oxidation by 8-hydroxygeraniol oxidase (8HGO) to yield 8-oxo-geranial[10–12] (Fig. 1a and Supplementary Fig. 1). The short-chain

dehydrogenase iridoid synthase (ISY) catalyses a 1,4 reduction of 8-oxo-geranial to form 8-oxocitronellyl enol, which then cyclizes to form 7S/R, 4aS and 7aR-nepetalactol, along with a number of side products[13–15]. Biosynthetic steps downstream of nepetalactol have been characterized in several plants, including *Catharanthus roseus* and *Camptotheca acuminata* (Supplementary Fig. 1)[12,16–19].

Of all possible nepetalactol stereoisomers, 7S-cis-trans (7S, 4aS, 7aR) and 7R-cis-cis (7R, 4aS, 7aR) nepetalactol are most commonly observed in nature (Fig. 1a)[1]. 7S-cis-trans nepetalactol-derived ('route I') iridoids, which are also precursors for alkaloid biosynthesis, are found among diverse asterid orders, whereas 7R-cis-cis nepetalactol-derived ('route II') iridoids primarily occur in Lamiales families[1,20,21]. The stereo-configuration at the C-7 position is set by a lineage-specific stereoselective ISY that generates either S- or R-8-oxocitronellyl enol (catalysed by 7S-ISY and 7R-ISY, respectively)[13,15]. Although 7S-cis-trans nepetalactol

[1]Department of Natural Product Biosynthesis, Max Planck Institute for Chemical Ecology, Jena, Germany. [2]Center for Applied Genetic Technologies, University of Georgia, Athens, GA, USA. [3]Department of Crop and Soil Sciences, University of Georgia, Athens, GA, USA. [4]Future Genomics Technologies, Leiden, The Netherlands. [5]Universidade de Ribeirão Preto, Departamento de Biotecnologia, Ribeirão Preto, Brazil. [6]Institute of Plant Breeding, Genetics and Genomics, University of Georgia, Athens, GA, USA. [7]The Plant Center, University of Georgia, Athens, GA, USA. ✉e-mail: mmartinez@ice.mpg.de; Robin.Buell@uga.edu; oconnor@ice.mpg.de

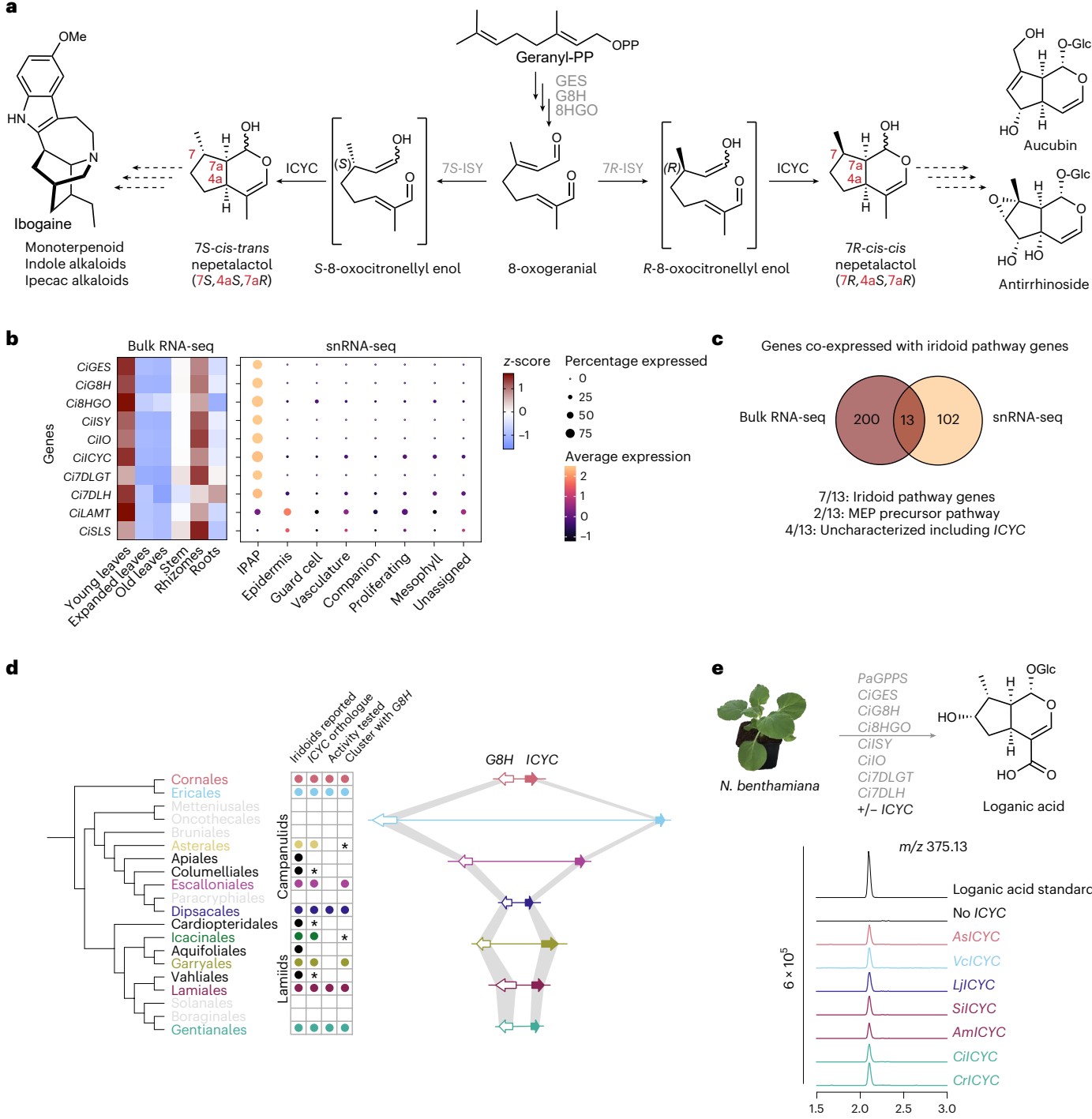

**Fig. 1 | Identification of ICYC. a**, Scheme showing iridoid biosynthesis including previously characterized iridoid pathway genes (grey). The 8-oxogeranial intermediate can be converted to either 7S-cis-trans nepetalactol or 7R-cis-cis nepetalactol by previously characterized species-specific stereoselective ISYs together with the newly identified ICYCs. For a complete scheme of the secoiridoid pathway including all intermediates, see Supplementary Fig. 1. **b**, *C. ipecacuanha* bulk tissue RNA-seq and snRNA-seq show tight co-expression of orthologues of previously characterized secoiridoid pathway genes from *C. roseus* (Supplementary Fig. 3a) and the newly identified *CiICYC*. Complete tissue-specific data are shown in Supplementary Fig. 3b. Cell clusters were grouped into cell types that were determined using marker genes (see Supplementary Fig. 4 for complete analysis of snRNA-seq dataset). **c**, Venn diagram showing overlay of co-expressed genes from both datasets. See Extended Data Fig. 1a for a detailed list of genes. MEP, 2-C-methyl-D-erythritol 4-phosphate. **d**, Identification of *ICYC* orthologues

in various iridoid-producing orders. The tree is based on the latest angiosperm phylogeny[35]. Circles depict reported presence of iridoids within at least one species of the respective order. An asterisk denotes that no sequencing data were publicly available from a reported iridoid-producing species from the respective clade, and thus *ICYC* presence could not be determined. Right: a scheme depicting the synteny of the *ICYC* and *G8H* gene cluster. The synteny is shown for representative species of each order: *A. salviifolium* (Cornales), *V. corymbosum* (Ericales), *Escallonia rubra* (Escalloniales), *L. japonica* (Dipsacales), *Eucommia ulmoides* (Garryales), *A. majus* (Lamiales) and *C. ipecacuanha* (Gentianales)[100–104]. **e**, Representative *ICYC* orthologues from different orders all reconstituted the iridoid pathway up to loganic acid in *N. benthamiana*. *Ci*, *C. ipecacuanha*; IO, iridoid oxidase; 7DLH, 7-deoxyloganic acid hydroxylase; LAMT, loganic acid methyltransferase; SLS, secologanin synthase; *As*, *A. salviifolium*; *Vc*, *V. corymbosum*; *Lj*, *L. japonica*; *Si*, *S. indicum*; *Am*, *A. majus*; *Ci*, *C. ipecacuanha*; *Cr*, *C. roseus*.

is observed in low yields when 7S-ISY is incubated with 8-oxogeranial, it has long been hypothesized that an additional enzyme assists the cyclization of 3S- or 3R-8-oxocitronellyl enol to 7S-cis-trans and 7R-cis-cis nepetalactol for the following reasons: (a) spontaneous cyclization occurs only in vitro and almost exclusively yields the cis-trans configuration, thus failing to account for the existence of the 7R-cis-cis configuration when 7R-ISY is used with 8-oxogeranial[15]; (b) the known biosynthetic enzymes (GES, G8H, 8HGO, ISY) are not sufficient to reconstitute nepetalactol biosynthesis in *Nicotiana benthamiana*[12,22]; and (c) recent work on *Nepeta*, a Lamiaceae genus that independently evolved iridoid biosynthesis, identified *Nepeta*-specific proteins that assist formation of the 7S-cis-trans nepetalactol scaffold from S-8-oxocitronellyl enol[14,23,24]. Indeed, the inclusion of a *Nepeta* Major Latex Protein Like (MLPL) enabled the successful pathway reconstitution of 7S-iridoids and downstream alkaloids in yeast and *N. benthamiana*[22,25].

Identifying the asterid iridoid cyclase(s) (ICYCs) responsible for formation of 7S-cis-trans and 7R-cis-cis nepetalactol has been a long-standing challenge. The cyclization of this enol species to form the bicyclic scaffold that characterizes nepetalactol is an unusual chemical transformation that could be catalysed by any number of protein scaffolds. To obtain a refined pool of gene candidates for this biosynthetic step, we generated de novo genome assemblies and high-resolution expression data of two evolutionarily distant members of the asterid clade[26], *Alangium salviifolium* (Cornales) and *Carapichea ipecacuanha* (Gentianales) along with single-nuclei transcriptomics of *C. ipecacuanha* (Supplementary Fig. 2–5 and Supplementary Tables 1–7). We identified highly conserved orthologues of *C. roseus* secoiridoid pathway genes in both species, confirming that iridoid biosynthesis is ancestral to asterids (Supplementary Fig. 3a). We next performed co-expression analysis using tissue-specific RNA-sequencing (RNA-seq) data and found iridoid pathway genes to be tightly co-expressed in *C. ipecacuanha* young leaves and rhizomes and in *A. salviifolium* roots (Fig. 1b and Supplementary Fig. 3b,c). In addition, we constructed gene co-expression networks on the cell clusters obtained from the single-nuclei RNA-seq (snRNA-seq) data of *C. ipecacuanha* young leaves (Fig. 1b, Supplementary Figs. 4 and 5 and Supplementary Dataset 1). Iridoid pathway orthologues up to 7-deoxyloganic acid hydroxylase (*7DLH*) are tightly co-expressed in a cell cluster corresponding to the internal phloem associated parenchyma (IPAP) cells, a cell type that has been previously shown to harbour early and intermediate iridoid biosynthesis steps in *C. roseus*[27]; later iridoid pathway gene orthologues are expressed in cell clusters containing epidermis marker genes (Fig. 1b and Supplementary Fig. 5b). The observed compartmentalization of IPAP-specific iridoid biosynthesis and epidermis-specific downstream biosynthesis is shown here for a *Rubiaceae* species and is identical to that found in *C. roseus* (*Apocynaceae*), suggesting that this spatial organization is conserved between these iridoid-producing families[12,27–30].

With these datasets in hand, we filtered the transcript lists generated from bulk tissue RNA-seq and snRNA-seq co-expression analyses for high absolute expression levels (counts per million (CPM) > 50 in young leaves; cluster average expression >1 within the IPAP cell cluster), based on the assumption that the cyclase gene would be highly expressed. Combining the filtered lists provided 13 gene candidates (Fig. 1c and Extended Data Fig. 1a). These 13 genes included all seven IPAP-specific iridoid biosynthesis genes, the two rate-limiting genes *DXS* and *DXR* of the 2-C-methyl-D-erythritol 4-phosphate pathway that makes the GPP precursor isopentenyl phosphate, and only four uncharacterized genes (Fig. 1c and Extended Data Fig. 1a). It is worth noting that when less stringent absolute expression value cut-offs were applied, the overlaid list also contained orthologues of the three known *C. roseus* basic helix–loop–helix iridoid biosynthesis transcriptional regulators, highlighting that this strategy can also be used to identify cell-type-specific transcriptional regulators (Extended Data Fig. 1b)[31–33].

We had previously developed a mass spectrometry-based detection method for 7S-cis-trans nepetalactol-derived iridoids in transfected *N. benthamiana* leaves[22]. We capitalized on this method to screen the activity of the cyclase gene candidates by assaying them in the context of upstream (*GPPS*, *GES*, *G8H*, *8HGO*, *ISY*) and downstream (7-deoxyloganetic acid glucosyl transferase (*7DLGT*), *7DLH*)) iridoid biosynthetic pathway genes. The cyclase candidates were expressed with *C. ipecacuanha* orthologues of iridoid biosynthetic genes predicted to generate loganic acid, an iridoid derived from 7S-cis-trans nepetalactol, in *N. benthamiana*. Inclusion of one of the candidates[34], which was functionally annotated as a MES, resulted in the efficient production of loganic acid (Fig. 1e). The enzyme was thus named Iridoid cyclase (ICYC). This protein is entirely unrelated to the *Nepeta*-specific cyclases (nepetalactol-related short chain reductases and MLPLs), clearly indicating that iridoid cyclization arose convergently at least once. We systematically compared the *C. ipecacuanha* ICYC (CiICYC) sequence against available data for 20 asterid clades[35] and identified ICYC orthologues in all clades reported to produce iridoids with the exception of the Apiales and Aquifoliales iridoid-producing genera *Griselinia* and *Helwingia*, respectively[36,37]. Analogously, an ICYC orthologue appeared to be absent in non-iridoid-producing clades, as well as in Nepeta (Fig. 1d; see Supplementary Fig. 6 for a tree with all orthologues). It is worth noting that *ICYC* is located next to the iridoid pathway gene *G8H*, forming a small biosynthetic gene cluster that is conserved in all iridoid-producing orders for which genome data is available (Fig. 1d).

In addition to CiICYC (lamiids, Gentianales, Rubiaceae), we selected orthologues from six additional species representing different asterid orders and families: the 7S-cis-trans nepetalactol-producing species *A. salviifolium* (AsICYC, Cornales), *Vaccinium corymbosum* blueberry (VcICYC, Ericales), *Lonicera japonica* (LjICYC, campanulids, Dipsacales) and *C. roseus* (CrICYC, lamiids, Gentianales, Aponcynaceae), and the 7R-cis-cis nepetalactol-producing Lamiales species *Sesamum indicum* (SiICYC, Pedaliaceae) and *Antirrhinum majus* (AmICYC, Plantaginaceae)[15,26,38–40]. Inclusion of each of these genes resulted in the effective reconstitution of loganic acid biosynthesis in *N. benthamiana* (Fig. 1e and Supplementary Fig. 7). We further used these cyclases to reconstitute the complete downstream *C. ipecacuanha* and *A. salviifolium* iridoid pathways (Supplementary Fig. 8). Finally, to confirm ICYC activity in a native plant, we performed virus-induced gene silencing (VIGS) of CrICYC in *C. roseus* (a technique not available in *C. ipecacuanha*) and detected reduced iridoid (secologanin) content, consistent with the proposed iridoid cyclization function (Extended Data Fig. 2).

To rigorously characterize the direct product of ICYC, we assayed seven recombinantly produced ICYC orthologues together with 8-oxogeranial and the previously characterized 7S-ISY from *C. roseus* (CrISY) and 7R-ISY from *A. majus* (AmISY) (Fig. 2 and Supplementary Figs. 9 and 10)[13,15]. CrISY and AmISY reduce 8-oxogeranial to form the unstable intermediate S- or R-8-oxocitronellyl enol, respectively, which, in the absence of a cyclase, spontaneously forms a small amount of cis-trans nepetalactol, along with a wide variety of side products (Fig. 2a; see Supplementary Fig. 9 for detailed description of all spontaneous products)[14,15]. When CrISY was assayed together with any of the ICYC orthologues, 7S-cis-trans (7S, 4aS, 7aS) nepetalactol accumulation was substantially increased compared to when CrISY was assayed alone (Fig. 2b). When AmISY was assayed with ICYC orthologues, a new peak appeared (Fig. 2c). We confirmed the identity of this ICYC-specific product as 7R-cis-cis (7S, 4aS, 7aS) nepetalactol by chemical oxidation and comparison with an authentic standard of 7R-cis-cis nepetalactone (Fig. 2d–f)[23]. In addition, we observed comparable results when we assayed the ICYC orthologues directly with S-8-oxocitronellal and R-8-oxocitronellal at high concentrations of general acid (0.5 M 3-(N-morpholino)propanesulfonic acid (MOPS)), conditions that promote partial tautomerization to the putative cyclase substrate, 8-oxocitronellyl enol[14] (Supplementary Fig. 11).

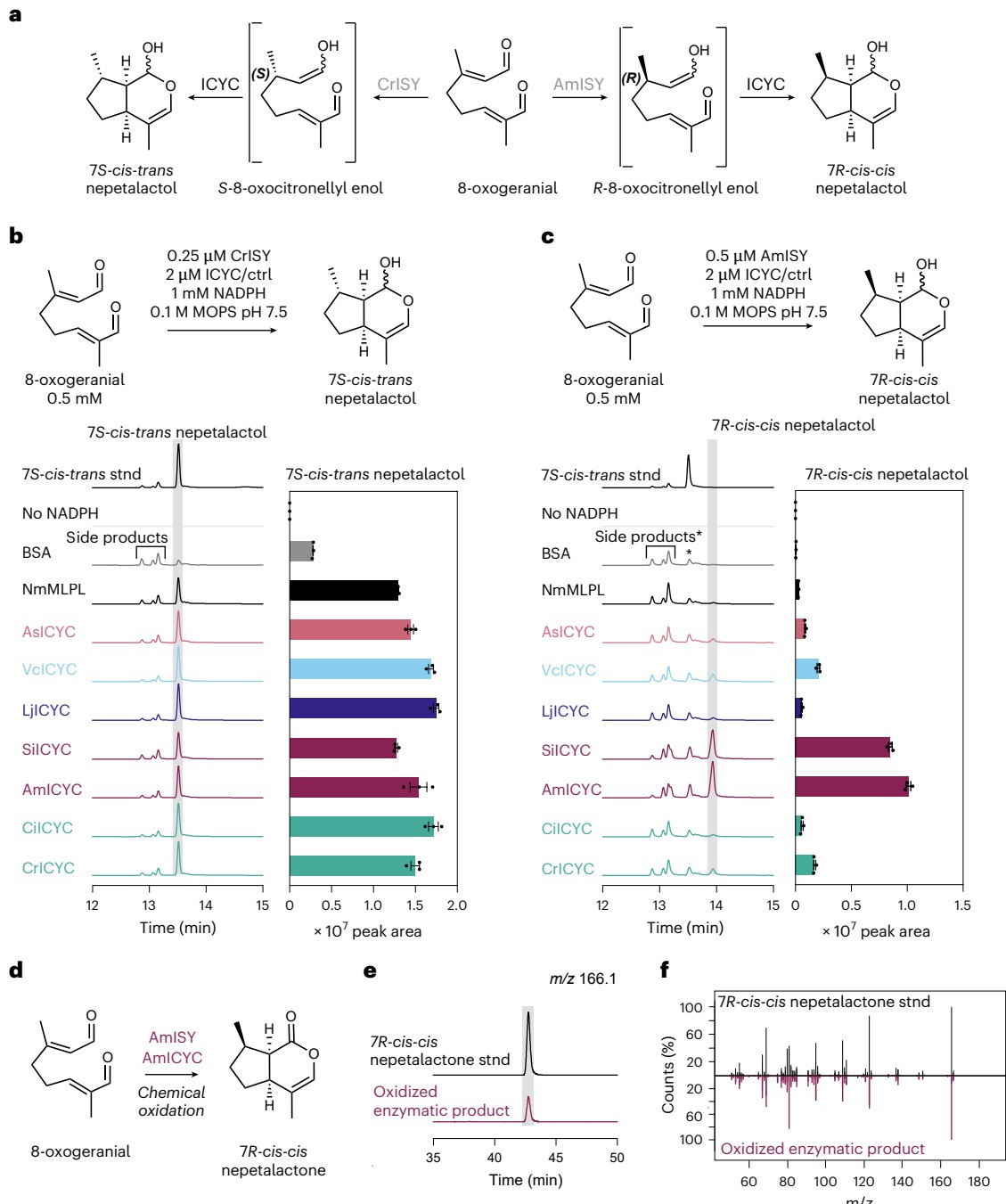

**Fig. 2 | In vitro activity assays reveal stereoselectivity of ICYCs. a**, Scheme depicting reaction products of 7*S* and 7*R* stereo-selective ISY together with ICYC. **b,c**, Assays were performed as indicated and analysed by GC-MS. BSA was used as a negative control. Displayed chromatograms are total ion chromatograms of one representative replicate. See Supplementary Fig. 9 for detailed descriptions of chromatograms including side products and Supplementary Fig. 10 for fragmentation patterns of standards and enzymatic products. Bar graphs depict peak areas from $N = 3$ replicates of nepetalactol (shaded in grey); error bars are standard error of the mean (s.e.m.). **b**, Assays of ICYC orthologues with the 7*S* selective *C. roseus* ISY (CrISY) show a dramatic increase in 7*S-cis-trans* nepetalactol in the presence of ICYC orthologues. **c**, Assays of ICYC orthologues with the 7*R* selective *A. majus* ISY (AmISY) reveal appearance of a peak consistent

with 7*R-cis-cis* nepetalactol in the presence of ICYC. The asterisk indicates that these side products are the 7*R* enantiomers that coelute with the 7*S* series when using an achiral stationary phase as used here (Supplementary Fig. 9). **d**, To confirm the identity of the AmISY–AmICYC enzymatic product, five enzymatic reactions were pooled and chemically oxidized to 7*R-cis-cis* nepetalactone (Methods). **e**, Chiral GC-MS analysis of the chemically oxidized product showed that it had the same retention time and the identical mass (extracted ion chromatogram for nepetalactone mass $m/z$ 166.1) as the authentic standard[23] on a chiral GC column. **f**, The MS fragmentation pattern of the oxidized enzymatic product was identical to the standard confirming that the enzymatic product is indeed 7*R-cis-cis* nepetalactol. Ctrl, control.

Taken together, these results show that ICYCs exclusively produce 4a*S*, 7a*R* nepetalactol regardless of the stereochemistry of the C-7-position. Strikingly, while all ICYC orthologues generate *7S-cis-trans* nepetalactol from *S*-8-oxocitronellal, *7R-cis-cis* nepetalactol is most efficiently

generated when ICYC orthologues from the Lamiales species known to produce this stereoisomer (that is, AmICYC and SiICYC) are used. This observation suggests that enzymes in these species might have specialized to adapt to the 7*R* substrate. We also assayed the

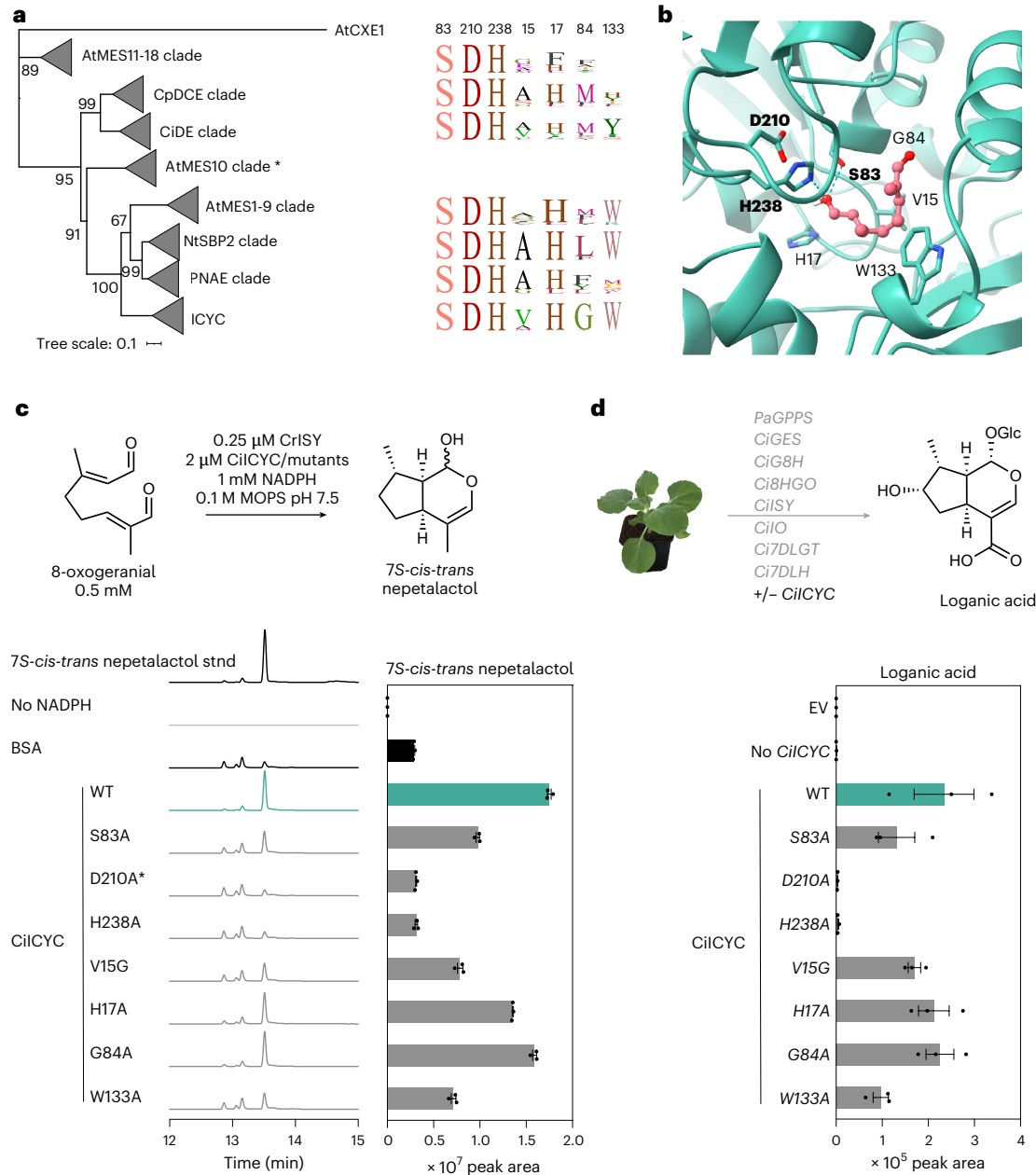

**Fig. 3 | ICYC is a member of the MES family. a**, A condensed phylogenetic tree showing that ICYC is related to MESs that esterify substrates in plant hormone activation pathways and in alkaloid biosynthesis. Each clade is named after a member found within the respective clade. See Supplementary Fig. 12 for an extended version of this tree. Sequence data highlight the conservation of the catalytic triad and other amino acid residues that were subjected to mutation in CiICYC. The asterisk indicates that AtMES10 was a single-member clade, and thus no sequence logos were generated. At, *Arabidopsis thaliana*; CXE, carboxylesterases; CiDE, *C. ipecacuanha* deacetyl(iso)ipecoside esterase; CpDCE, *Cinchona pubescens* dihydrocorynantheine aldehyde esterase; RsPNAE, *Rauvolfia serpentina* polyneuridine aldehyde esterase. **b**, Putative active site of an alphafold3 structural model of CiICYC with the docked (AutoDock Vina) substrate *S*-8-oxocitronellyl enol substrate. Indicated amino acids were chosen for mutagenesis studies because they are part of the catalytic triad (in bold) or are in proximity to the docked substrate and are conserved in ICYC orthologues. **c**, In vitro assays show that D210A and H238A ICYC mutants yield a reaction profile that is identical to the negative control. S83A, V15G and W133A mutants show reduced levels of nepetalactol compared to wild-type ICYC, whereas H17A and G84A mutants behaved similarly to wild-type (WT) ICYC. The asterisk indicates that the D210A mutant exhibited poor solubility, and thus the assays with this mutant should be interpreted cautiously. Bar graphs depict normalized peak areas from $N = 3$ replicates of nepetalactol (shaded in grey); error bars are s.e.m. **d**, Reconstitution of loganic acid biosynthesis in *N. benthamiana* with CiICYC and CiICYC mutants yielded results that are consistent with results from in vitro assays shown in **c**. Bar graphs depict normalized peak areas from $N = 3$ biological replicates; error bars are s.e.m. EV, empty vector.

product profile when ICYC was added after ISY reduced 8-oxogeranial to 8-oxocitronellyl enol (Extended Data Fig. 3a). Under these conditions, no increase in nepetalactol is observed, suggesting that 8-oxocitronellyl enol must be immediately transferred to ICYC after it is formed by ISY. In support of this hypothesis, we found that ISY and ICYC interact (as measured by split luciferase assays) (Extended Data Fig. 3b).

However, this observation does not definitively show that substrate channelling, which would protect the reactive enol species, in fact takes place.

ICYC is a methyl esterase (MES)-type α/β hydrolase. These MESs are ubiquitously found in plants but have not previously been associated with non-esterase functions such as cyclization. Many MES

family members play roles in activating the plant hormones methyl jasmonate, methyl salicylate or indole-3-acetic acid methyl ester via demethylation[41–43]. Other members act as esterases in monoterpene indole or ipecac alkaloid biosynthetic pathways[26,44–48]. All α/β hydrolases use a conserved catalytic triad (serine, aspartate/glutamate and histidine) along with a glycine-rich oxyanion hole to catalyse substrate hydrolysis (Fig. 3a,b)[49,50]. Although ICYC orthologues form a phylogenetic clade well separated from all other MESs, indicating a monophyletic origin (Fig. 3a and Supplementary Fig. 12), the catalytic triad and oxyanion hole motifs are still present, as in canonical esterases (Fig. 3a and Supplementary Fig. 13). No obvious differences were noted within the active sites of ICYCs from species that produce 7R nepetalactol compared to those that produce 7S nepetalactol (Supplementary Fig. 14). All ICYC orthologues showed esterase activity when tested with the model substrate 4-nitrophenyl acetate, indicating that these catalytic motifs remain functional (Extended Data Fig. 4). To assess which residues are responsible for catalysis of cyclization, we docked the S-8-oxocitronellyl enol substrate to a CiICYC alphafold3 model with AutoDock Vina to pinpoint residues in the binding pocket. We also used our phylogenetic analysis to select amino acid residues that are conserved in ICYC orthologues (Fig. 3a,b). Unsurprisingly, mutations of any of the catalytic triad amino acid in CiICYC (S83A, D210A, H238A) abolished esterase activity (Extended Data Fig. 4c)[51]. D210A was largely insoluble and could therefore not be assayed reliably. When these mutants were tested for cyclase activity in vitro (Fig. 3c) and in N. benthamiana (Fig. 3d), D210A and H238A mutants were inactive, whereas the S83A mutant exhibited some cyclization activity. Out of the four additional mutants within the putative cyclase active site, V15G and W133A led to reduced amounts of the nepetalactol product (Fig. 3c,d). Although more extensive experimentation is required to understand the mechanism of this unusual cyclization (Supplementary Fig. 15), we speculate that these residues shape the binding pocket of ICYC, allowing 8-oxocitronellyl enol to adopt the conformation required for cyclization. The amino acid H238, the mutation of which abolished detectable activity, may play a role in orienting the substrate. Alternatively, this residue could interact with the enol moiety of 8-oxocitronellyl enol, which could in turn activate the substrate to undergo stepwise cyclization (Supplementary Fig. 15d)[52].

The discovery of these stereoselective ICYC orthologues fully explains the dominance of 7S cis-trans and 7R cis-cis nepetalactol derived iridoids in the asterids. This work clearly shows that ICYC was recruited from widely present MESs to perform this unusual cyclization reaction and has since been evolutionary conserved throughout iridoid-producing asterids. It is worth noting that an apparent evolutionary loss of ICYC in the Nepetoideae may have facilitated the convergent evolution of cyclases with different stereoselectivities in Nepeta. The discovery of ICYC is an excellent starting point to mechanistically understand this unusual cyclization reaction[52,53]. Finally, this discovery also unlocks the possibility for formation of previously inaccessible iridoid stereoisomers, which will enable metabolic engineering for sustainable production of valuable iridoid and iridoid-derived compounds.

## Methods

### Plant sampling and RNA extraction

C. ipecacuanha and A. salviifolium plants were previously obtained[26]. Plants labelled with different numbers refer to independent individual plants. RNA-seq data from plants labelled as 'plant 1' was previously published (National Center for Biotechnology Information (NCBI) BioProject PRJNA1169657). All plants were grown under the following conditions: 12/12 h light/dark, 28–30 °C/24–26 °C temperature, 70–80% humidity. Plants for Illumina sequencing and expression analyses (see Supplementary Fig. 2a,b for photographs) were the following: C. ipecacuanha plants labelled as 'plant 2' and 'plant 3' were both 1.5 years old; A. salviifolium plants labelled as 'plant 2' and 'plant 3'

were 2.5 and 4 years old, respectively. Dissected tissues from C. ipecacuanha plants 2 and 3 and A. salviifolium plant 2 were flash frozen and shipped on dry ice to FutureGenomics for RNA extraction and sequencing (see below). Additional tissues were collected for Oxford Nanopore Technologies (ONT) sequencing for genome annotation (young leaves, mature leaves, green stem and roots of 2-year-old C. ipecacuanha plants; leaf buds, young leaves, mature leaves and roots of 3-year-old A. salviifolium plants). RNA of samples for ONT sequencing and A. salviifolium plant 3 samples was extracted in-house as follows. Dissected tissues were immediately flash frozen into liquid nitrogen and ground in liquid nitrogen using an IKA A11 basic analytical mill or mortar and pestle. Total RNA was extracted using the RNeasy Plant Mini Kit (Qiagen) according to the manufacturer's instructions, including on column DNAse digest. RNA concentrations and purity were determined with a Nanophotometer N60 (Implen).

### RNA Illumina sequencing

Illumina sequencing of C. ipecacuanha plants 2 and 3 and A. salviifolium plant 2 was performed at FutureGenomics as follows. Flash-frozen plant tissues were powdered in liquid nitrogen using mortar and pestle, and RNA was extracted using Zymo Quick-RNA Plant Kit (Zymo Research). The quality of the RNA was analysed using RNA ScreenTape on an Agilent 4200 TapeStation System (Agilent Technologies Netherlands BV), and the quantity was measured using a Qubit 3.0 Fluorometer (Life Technologies Europe BV). Illumina RNAseq libraries were prepared using NEBNext Ultra II Directional RNA Library Prep Kit Illumina and NEBNext Multiplex Oligos for Illumina (New England Biolabs). The quality of the libraries was checked using Agilent D1000 screentape on an Agilent 4200 TapeStation System. RNA library paired-end sequencing (2 × 150 bp) was performed using Illumina's NovaSeq 6000 technology. Two rounds of sequencing were carried out to reach the desired output target.

Illumina sequencing of A. salviifolium plant 3 was performed at Novogene. RNA samples were shipped to Novogene where messenger RNA library preparation and sequencing were performed according to the company's standard protocol for mRNA sequencing. RNA integrity and quantitation were assessed on a Bioanalyzer (Agilent Technologies). All samples were above the required minimum RNA integrity number value. Sequencing was performed on an Illumina NovaSeq X Plus PE150 platform with a data output target of 9 G of raw data.

### ONT full-length complementary DNA sequencing

mRNA was purified from the total RNA using the Dynbeads mRNA Purification Kit (Invitrogen) and input into the ONT SQK-PCS111 kit to generate full length complementary DNA libraries. Resultant libraries were sequenced on a MIN106 Rev. D flowcell before being basecalled using Guppy (v6.5.7) (https://nanoporetech.com/software/other/guppy) using the super high accuracy model (dna_r9.4.1_450bps_sup.cfg) and the parameters -q 0, –trim_strategy none and --calib-detect.

### Genome sequencing and assembly

Nuclei were extracted from young leaves using a nuclei isolation protocol[54]. The nuclear pellet was resuspended in Qiagen buffer G2 with RNaseA and proteinaseK, and DNA extraction was further continued according to the instructions of the Qiagen Genomic Tip/100 G protocol (Qiagen). The quality of the DNA was analysed using Genomic DNA ScreenTape on an Agilent 4200 TapeStation System (Agilent Technologies Netherlands BV), and the quantity was measured using a Qubit 3.0 Fluorometer (Life Technologies Europe BV). Nanopore sequencing libraries were prepared using the Ligation Sequencing Kit V14 (SQK-LSK114) according to the manufacturer's instructions (ONT). Each library was run on an R10.4.1 PromethION flowcell (FLO-PRO114M; ONT) and reloaded on a daily basis after a nuclease flush with Flow Cell wash kit (EXP-WSH004). Super-accuracy base calling was done using Guppy 6.3.2. Illumina DNAseq libraries were prepared using the

Nextera Flex kit according to the manufacturer's instructions (Illumina) and were sequenced in paired-end mode (2 × 150 bp) using Illumina's NovaSeq 6000 technology.

ONT genomic reads were assembled using Flye (v.2.9.1), with settings 'overlap 10K, error rate 0.025, no-alt-contigs'[55]. The genome contigs were polished using nanopore reads by Medaka (https://github.com/nanoporetech/medaka) and then polished twice using Illumina reads by Pilon (v.1.23)[56]. The polished genome sequence was then collapsed using Purge dups v1.2.6[57].

### Genome annotation

The genome assemblies were repeat masked by first creating a custom repeat library (CRL) for each genome. Repeats were first identified with RepeatModeler[58] (v2.03), and protein-coding genes were filtered out from the repeat database using ProtExcluder[59] (v1.2) to create a CRL. The CRL was then combined with Viridiplantae repeats from RepBase (v20150807) to generate the final CRL for each genome. Each genome assembly was repeat-masked using the respective final CRL and RepeatMasker[60] (v4.1.2-p1) using the parameters -e ncbi -s -nolow -no_is -gff.

RNA-seq libraries were processed for genome annotation by first cleaning with Cutadapt[61] (v2.10) using a minimum length of 100 nt and quality cut-off of 10 then aligning the cleaned reads to the respective genome using HISAT2[62] (2.1.0). ONT cDNA reads were processed with Pychopper (https://github.com/epi2me-labs/pychopper) (v2.7.10), and trimmed reads greater than 500 nt were aligned to the respective genome using minimap2[63] (v2.17-r941) with a maximum intron length of 5,000 nt. The aligned RNA-seq and ONT cDNA reads were each assembled using Stringtie[64] (v2.2.1), and transcripts less than 500 nt were removed.

The initial gene models for each genome were created using BRAKER2[65] (v2.1.6) using the soft-masked genome assemblies and the aligned RNA-seq libraries as hints. The gene models were then refined using two rounds of PASA2[66] (v2.5.2) to create a working gene model set for each genome. High-confidence gene models were identified from each working gene model by filtering out gene models without expression evidence or a PFAM domain match, or were a partial gene model or contained an interior stop codon. Functional annotation was assigned by searching the working gene models proteins against The *Arabidopsis* Information Resource (TAIR)[67] (v10) database and the Swiss-Prot plant proteins (release 2015_08) database using BLASTP[68] (v2.12.0) and the PFAM[69] (v35.0) database using PfamScan[70] (v1.6) and assigning the annotation based on the first significant hit.

### Single-nuclei RNA-seq of *C. ipecacuanha* young leaves

Nuclei were isolated following the protocol outlined by ref. 71. For *C. ipecacuahna* young leaves 0.01% Triton X-100 was used, and RNase inhibitor (Sigma Protector RNase Inhibitor, catalogue number 3335402001; SigmaAldrich) was added to the nuclei isolation buffer for a final concentration of 0.5 U µl⁻¹. Nuclei were stained with DAPI (4′,6-diamidino-2-phenylindole) and sorted using fluorescence activated cell sorting. Nuclei were concentrated by spinning at 300 *g* for 5 min. Concentrated nuclei were used for single-cell RNA-seq library construction using the PIPseq T20 v4.0Plus Kit (Illumina) with 1 µl of additional RNase inhibitor.

### Single-cell transcriptomics and co-expression analyses

Single-nuclei RNA-seq reads were processed using the 'barcode' command from pipseeker-v3.1.3 (Illumina). The pipseeker processed fastq files, and the generated barcode whitelist was used as input into the STARsolo (v2.7.10b)[72] alignment program. The following parameters were used: --alignIntronMax 5000, --soloUMIlen 12, --soloCellFilter EmptyyDrops_CR, --soloFeatures GeneFull, --soloMultiMappers EM and --soloType CB_UMI_Simple. Seurat v4.3.0.1 was used for downstream analysis. Samples were filtered to retain high-quality cells by removing cells with less than 300 genes or more than 10,000 genes and less than

500 UMIs (unique molecular identifiers) or more than 30,000 UMIs. Samples were also run through DoubletFinder v2.0.4[73] to remove suspected doublets. Reciprocal principal component analysis was used to integrate the two replicates together using the top 3,000 variable genes. The top 60 principal components were used with a resolution parameter of 0.5 to calculate the uniform manifold approximation and projection.

SimpleTidy_GeneCoEx v2.0.0[74] was used for co-expression analysis. An *R* value cut-off of 0.7 with a resolution parameter of 3 was used to generate 27 modules containing 5 or more genes, which comprised 29,765 genes total. Module 16 contained IPAP-specific genes of interest (1,496 genes in the module).

### Mapping of bulk RNA-seq to *A. salviifolium* and *C. ipecacuanha* assembled genomes

Adapter-cleaved raw fastq files were received, and reads were quality checked with FastQC (Galaxy version 0.73) and trimmed using Trimmomatic (Galaxy version 0.38.1)[75] on an in-house Galaxy server[76]. Reads belonging to the same sample but obtained in two sequencing rounds were concatenated. Reads were mapped to genomes using CLC Genomics workbench 21.0.4 (Qiagen) with these parameters: mismatch cost, 2; insertion cost, 3; deletion cost, 3; length fraction, 0.85; similarity fraction, 0.9; auto-detect paired distances, on; and maximum number of hits for a read, 20. Expression values are unique counts per gene. trimmed mean of M values normalized CPM values were used for downstream analyses. To identify iridoid pathway genes in *C. ipecacuanha* and *A. salviifolium*, amino acid sequences of the characterized iridoid pathway enzymes from *C. roseus* were blasted (tblastn) against the transcript working models derived from the genomes. The highest blast hit for each gene was chosen for expression analysis and cloning (Supplementary Fig. 3a). Blasting of enzymes known to perform cyclizations in *Nepeta*[14,24] yielded comparably poorly conserved sequences (maximum 59% identity) that were not co-expressed with iridoid pathway genes and were thus not considered as orthologues. *C. ipecacuanh*a tissue-specific co-expression analysis was performed on an expression atlas containing data from all tissues from plants 1–3. *CiISY* expression was used as a bait for Pearson correlation.

### Gene cloning

Complementary DNA was prepared from total RNA of *A. salviifolium* leaf buds and roots, *C. ipecacuanha* young leaves and *C. roseus* leaves (extracted as described above) using the iscript cDNA Synthesis Kit (Biorad) according to manufacturer's instructions. *NmMLPL* and *PaGPPS* genes had been cloned previously[22,24]. Genes from all other species and *CiICYC* protein mutants were obtained as synthetic sequences from Twist Biosciences. *CDS* sequences were amplified with the Q5 High-Fidelity 2X Master Mix (New England Biolabs) using cDNA or synthetic fragments as templates and gene-specific primers containing overhangs for In-Phusion cloning (Supplementary Table 8). Amplified sequences were gel-purified using the Zymoclean Gel DNA Recovery Kit (Zymo Research) and cloned using the 5x In-Fusion Snap Assembly Master Mix (TaKaRa Bio). For pathway reconstitution in *N. benthamiana*, coding regions were inserted into a modified 3Ω1 vector (contains *UBQ10* promoter and terminator from *Solanum lycopersicum*[77]) previously digested with BsaI-HF v2 (New England Biolabs, NEB). For expression in *Escherichia coli*, sequences were cloned into the pOPINF vector[78] previously digested with KpnI-HF and HindIII-HF (NEB). ICYCs are predicted to be cytosolic and do not contain any transit peptides to be cleaved off for recombinant protein production (predicted by DeepLoc 2.1 (ref. 79)). For split-luciferase assays, CiICYC and CiISY were cloned into KpnI-HF and SalI-HF (NEB) digested pCAMBIA1300-NLuc (N-terminal luciferase) and KpnI-HF and PstI-HF (NEB) digested pCAMBIA1300-CLuc (C-terminal luciferase)[80]. Cloning reactions were transformed into heat shock competent *E. coli* TOP10 and grown overnight in a 37 °C incubator on Luria–Bertani

(LB) agar plates containing the respective antibiotics. Plasmids were isolated from overnight cultures of single colonies using the Wizard Plus SV Minipreps DNA Purification System kit (Promega), and inserted sequences were confirmed by Sanger sequencing.

### *Agrobacterium tumefaciens* mediated transient expression in *N. benthamiana*

*A. tumefaciens* GV3101 cells were transformed through electroporation, recovered in YEB without antibiotics and incubated on YEB plates containing antibiotics (rifampicin and gentamycin and the appropriate antibiotic for plasmid selection) at 28 °C for 48 h. Colony PCR was done on single colonies to confirm the presence of plasmids. Positive colonies were grown in liquid YEB for 24 h. From these cultures glycerol stocks were prepared and stored at –80 °C. Three- to four-week-old *N. benthamiana* plants (grown in a greenhouse with 16 h/8 h light/dark, 23–26 °C/16–22 °C temperature and 40–70% humidity) were used for agroinfiltration[26,81]: cells from glycerol stocks were spread on YEB plates containing antibiotics and 100 μM acetosyringone and grown for 24 h until a visible layer of bacteria appeared. The bacteria were transferred to 1–2 ml of infiltration medium (10 mM MES, 10 mM $MgCl_2$, 100 μM acetosyringone, pH 5.7) and gently resuspended, and the optical density at 600 nm ($OD_{600}$) was measured in 1:10 dilutions using an Implen OD600 DiluPhotometer. For pathway reconstitution experiments, strains were mixed and diluted in infiltration buffer to $OD_{600}$ = 0.1 per strain. For split-luciferase assays (see below), the strains were mixed at $OD_{600}$ = 0.13 per strain. Strains harbouring constructs with NLuc or CLuc fused to CiICYC of CiISY or empty vectors containing free NLuc or CLuc as controls were infiltrated in combinations as indicated. A strain harbouring a construct with the *p19* gene was co-infiltrated in all cases. The culture mixtures were infiltrated into *N. benthamiana* leaves and grown for 5 days under grow lights (16 h/8 h light/dark). Replicates are from individual plants. For metabolite analysis, leaf material was collected 5 days after agroinfiltration by flash freezing in tubes containing metal beads.

### Split-luciferase assays in *N. benthamiana*

Leaf disks (~1 cm) were punched from four biological replicates 3 days after infiltration and placed into a custom-made high-density polyethylene multi-well plate. The abaxial side of the leaf was facing up. Then 200 μl of 0.5 mM D-luciferin (Promega) was added to the leaf disks in each well. The plate was imaged with a NightSHADE LB 985 (Berthold Technologies) with luminescence emission at 0.1 s (wavelength filter 650 nm, 10% intensity) and 8 × 8 pixel binning. Images were taken after 15 and 20 min of incubation with luciferin in the dark with 10 s and 2 s exposure times, respectively. The pictures were analysed with indiGO™ 1.4 software (Berthold Technologies) and a scale from 5,000 to 65,000 counts per second was applied.

### VIGS in *C. roseus*

VIGS was performed according to an established method[29]. Briefly, 300 bp target regions of the coding regions of *C. roseus ICYC* and *ISY* genes (transcripts CRO_01G006740.1 and CRO_07G007680.1 at https://doi.org/10.5061/dryad.d2547d851, therein cro_v3.gene_models.cds.fa (https://datadryad.org/downloads/file_stream/2121644)) were selected using the SGN VIGS tool to avoid off-target gene silencing (https://vigs.solgenomics.net/)[82]. Genomic DNA was extracted from *C. roseus* leaves with the DNeasy Plant Mini Kit (Qiagen). Target region fragments were PCR-amplified from gDNA with in-fusion cloning overhangs using Phusion High-Fidelity DNA Polymerase (ThermoFisher) and specific primers (Supplementary Table 7). The obtained PCR fragments were cloned in the BamHI and XhoI (NEB) digested VIGS vector pTRV2-MgChl[83] using the In-Fusion Snap Assembly Master Mix (TaKaRa Bio).

*A. tumefaciens* GV3101 was transformed by electroporation as described above. For inoculation, *A. tumefaciens* GV3101 with plasmid pTRV1[84] and *A. tumefaciens* GV3101 carrying pTRV2-MgChl,

pTRV2-CrICYC or pTRV2-CrISY were grown overnight in a rotary shaker at 28 °C and 300 rpm in each 10 ml of LB medium supplemented with 50 mg l⁻¹ kanamycin, 25 mg l⁻¹ gentamicin and 100 mg l⁻¹ rifampicin to an $OD_{600}$ of ~2. Cultures were centrifuged for 10 min at 1,800 *g*, and pellets resuspended in infiltration buffer (100 μM acetosyringone, 10 mM NaCl and 1.75 mM $CaCl_2$) to an $OD_{600}$ of 2. After 2 h of incubation on a rotary shaker at room temperature and 60 rpm, 450 μl of each bacterial strain containing a pTRV2 plasmid were mixed with the same volume of the bacterial strain with pTRV1. VIGS inoculation was performed by pipetting 10 μl of the mixed bacterial suspension between plant stem and petiole of the first true leaf of a 30-day-old *C. roseus* cultivar 'Atlantis Burgundy Halo' plant (grown in a growth chamber at 16/8 h light/dark 23 °C/21 °C temperature, 50% humidity). The stem was pierced with a ø 0.40 × 25 mm Sterican needle (www.bbraun.com) twice through the bacterial suspension drop. Six plants per pTRV2 construct were inoculated. Plants inoculated with the pTRV2-MgChl strain served as controls. After VIGS inoculation, plants were grown under the same conditions as described above. Yellow tissues (due to co-silencing of *MAGNESIUM CHELATASE SUBUNIT H* gene) were collected 3 weeks after inoculation.

### Quantitative PCR

Total RNA of *C. roseus* silenced tissues was extracted using the RNeasy Plant Mini Kit (Qiagen) according to the manufacturer's instructions, including on column DNAse digest. RNA concentrations were measured with a Nanophotometer N60 (Implen). mRNA was reverse-transcribed with 0.5 μg total RNA as input using the iscript cDNA synthesis kit (Bio-rad) according to the manufacturer's instructions. Complementary DNA was then diluted 1:8 in water, and 2 μl of the dilution was used in each quantitative PCR (qPCR) reaction of 10 μl total volume in a 96-well plate with 333 nM of each primer (Supplementary Table 8) and 5 μl of PowerUp SYBR Green Master mix (Applied Biosystems) according to manufacturer's instructions. Amplification was done on a QuantStudio 1 (Applied Biosystems) qPCR machine with the following cycling conditions: 50 °C for 2 min, 95 °C for 2 min, followed by 40 cycles of 95 °C for 1 s and 60 °C for 30 s (ramp rate was 1.6 °C s⁻¹ in all cases). Amplification was followed by a melting curve (0.15 °C s⁻¹ from 60 °C to 95 °C) to confirm the presence of single amplification products. Data were analysed using the ΔΔCt method and normalized to the expression of the established reference gene *N2227*[85]. Graphs and statistics were done in Prism Graphpad 10.4.1.

### Metabolite extraction

Leaf material of agroinfiltrated *N. benthamiana* or silenced *C. roseus* was ground using two 4 mm metal beads and a TissueLyser (Qiagen) with pre-cooled adapters and extracted with 30 μl per mg of fresh material of 70% MeOH containing 0.1% formic acid and 1 μM harpagoside as internal standard. Samples were sonicated for 10 min, incubated on a rotator for 15 min and centrifuged at 18,000 *g* for 15 min. The supernatants were filtered through a 0.45 μm low binding hydrophilic PTFE filter plate (MultiScreen Solvinert 96, Merck-Millipore) into a 96-well Microtiter Plate (SureSTART WebSeal, Thermo Scientific) according to manufacturer's instructions. Plates were sealed with Rapid Slit Seal (BioChromato) and immediately analysed with ultra-performance liquid chromatography–tandem mass spectrometry (UPLC-MS/MS).

### Detection of iridoid glucosides by UPLC-MS/MS

The system consisted of an UltiMate 3000 Ultra-High Performance Liquid Chromatography system (Thermo Fisher Scientific) coupled to an Impact II high-resolution Quadrupole Time-Of-Flight mass spectrometer (Bruker Daltonics). A Kinetex XB C18 (2.1 × 100 mm, 2.6 μm; 100 Å) column (Phenomenex) was set at 40 °C and 0.6 ml min⁻¹ flow rate, and 2 μl of sample was injected. The mobile phase was A:B where A was water with 0.1% formic acid and B was acetonitrile. The gradient was as follows: 5% B at 0.5 min to 30% B at 6 min. Then the column

was flushed at 100% B until 7.6 min and re-equilibrated to 5% B until 10 min. Ionization was performed in negative electrospray ionization mode (ESI−) with 3,500 V capillary voltage and 500 V end plate offset, a nebulizer pressure of 2.5 bar with nitrogen at 250 °C and a flow of 11 l min$^{-1}$ as the drying gas. Acquisition was done at 12 Hz following a mass range from 100 to 1,000 $m/z$ with data-dependent MS/MS, an active exclusion window of 0.2 min and a reconsideration threshold of 1.8-fold change. Fragmentation was triggered on an absolute threshold of 400 and limited to a total cycle time range of 0.5 s. For collision energy, the stepping option model (from 20 to 50 eV) was used. Recalibration of the $m/z$ values took place at the start of each run using the expected cluster ion $m/z$ values of a direct source infusion of sodium formate-isopropanol solution. At the first minute of each run, the liquid chromatography (LC) input was redirected to waste. During this time, the $m/z$ values of the instrument were calibrated using the cluster ion $m/z$ values of a sodium formate-isopropanol solution injected by direct source infusion with a 5 ml syringe connected to an external pump at a flow rate of 0.18 ml h$^{-1}$. LC-MS data were collected with Bruker Compass qtofControl 5.2.109/Hystar 5.1.5.1 or Compass qtofControl 6.3 / Hystar 6.0.30.0 software.

## UPLC-MS/MS data analysis
Raw data were converted to mzml format using MSConvert 3.0.25036-69e37b6 and imported to MZmine 4.5.37 (ref. [86],[87]). Extracted ion chromatogram traces and MS/MS data of compounds of interest were exported from MZmine. Peak areas were calculated using the MZmine Processing Wizard and exported. Peak areas of compounds of interest were normalized to the internal standard harpagoside and converted to intensity per second. Further data analysis and construction of graphs was done in GraphPad Prism 10.4.1 for Mac OS X.

## Commercially available chemicals and standards
Secologanin (50741), loganic acid (PHL80492), 4-nitrophenyl acetate (N8130) and D-camphor (50843) were obtained from Sigma. NADPH tetrasodium salt (10621692001) was obtained from Roche; 8-oxogeranial (D476180) and 7$S$-$cis$-$trans$ nepetalactol (N390065) were purchased from Toronto Research Chemicals. Harpagoside (7471.2) was purchased from Roth.

## Synthesized standards and substrates
Secologanic acid standard was produced through alkaline hydrolysis of secologanin[26],[88]. Secologanin was incubated with 0.1 M NaOH (40 µl per 1 mg secologanin) for 5 h and then neutralized with HCl. The completeness of the reaction was confirmed through LC-MS analysis and the solution stored at −25 °C. $S$-8-Oxocitronellal, $R$-8-oxocitronellal and 7$R$-$cis$-$cis$ nepetalactone were previously obtained[13],[14],[23].

## Recombinant protein production and purification
Expression and purification were performed according to an established method with modifications[89]. $E.$ $coli$ SoluBL21 (DE3) was transformed by heat shock with pOPINF constructs. Pre-cultures were inoculated from single colonies, grown overnight at 37 °C and used to inoculate 100 ml 2× YT medium (500 ml in the case of CiICYC_D210A mutant). Cultures were grown at 37 °C until OD$_{600}$ 0.5–0.6 was reached, cooled to room temperature and induced with 0.2 mM isopropyl β-D-1-thiogalactopyranoside. After induction, the cultures were shifted to 18 °C over-night and collected the next day by centrifugation. Cell pellets were lysed on ice for 30 min with approximately 6 ml lysis buffer per 1 g of cell pellet (50 mM Tris–HCl pH8, 50 mM glycine, 5% glycerol, 500 mM NaCl, 20 mM imidazole, 0.2 mg ml$^{-1}$ lysozyme, 1 tablet per 50 ml of complete EDTA free protease inhibitor (Roche)) and sonicated for 2.5 min (2 s on, 3 s off) on ice (Bandelin UW 2070). Supernatants were incubated with gentle shaking in falcon tubes with 250 µl Ni-NTA Agarose (Qiagen) for 1 h at 4 °C to allow binding of His-tagged proteins. Slurry was pelleted gently by centrifugation at 1,000 $g$ for 30 s. The

supernatant was removed, and the slurry was washed three times with ice-cold wash buffer (50 mM Tris–HCL pH 8, 50 mM glycine, 5% glycerol, 500 mM NaCl, 20 mM imidazole) by inversion, centrifugation and removal of supernatant. Proteins were eluted using elution buffer (as wash buffer but containing 500 mM imidazole). Elution fractions were concentrated, and buffer was changed to storage buffer (20 mM HEPES, 150 mM NaCl, pH 7.5) using Amicon Ultra Centrifugal Filters (Millipore) with 10 kDa molecular weight cut-off according to manufacturer's instructions. Purity was assessed through SDS−PAGE, and concentration was determined using the extinction coefficient and by measuring the absorbance at 280 nm. Proteins were flash frozen in small aliquots in liquid nitrogen and stored at −70 °C.

## In vitro enzyme assays
With 8-oxogeranial as a substrate, enzymes were assayed under the following conditions: 100 mM MOPS pH 7.5, 0.5 mM 8-oxogeranial, 1 mM NADPH, 0.25 µM CrISY or 0.5 µM AmISY, and 2 µM ICYC, NmMLPL or bovine serum albumin (BSA) as control as indicated. The reactions were set up in a 100 µl total volume and started by the addition of NADPH. Reactions were incubated for 3 h at 30 °C and 400 rpm. For sequential reactions (Extended Data Fig. 3 ), ICYC, NmMLPL or BSA were added after 1.5 h of incubation with CrISY, and incubation was continued for an additional 1.5 h. For assays with $S$-, or $R$-8-oxocitronellal, conditions were as follows: 500 mM MOPS pH7.5, ~0.5 mM $S$-8-oxocitronellal or ~0.1 mM $R$-8-oxocitronellal, respectively, 5 µM of ICYC, NmMLPL or BSA. Reactions were incubated for 16 h at 30 °C and 400 rpm. After incubation, camphor (5 or 10 µl of a 1 mM solution in acetonitrile) was added as internal standard to each reaction and briefly vortexed. Reactions were then extracted with 200 µl ethyl acetate and vortexed thoroughly for 2 min. Layers were separated by centrifuging at 18,000$g$ for 5 min, and 100 µl of the upper ethyl acetate layer was transferred to a glass vial. Samples were immediately analysed by gas chromatography–mass spectrometry (GC-MS).

Esterase activities were measured in a spectrophotometric assay with 4-nitrophenyl acetate[90]. Reactions were performed in 96-well plates (CytoOne) at 250 µl total volume. Reactions contained 100 mM HEPES pH 7, 1 mM CaCl$_2$, 2.6 mM NaCl, 2 mM 4-nitrophenyl acetate and 0.5 µM of enzyme or BSA as control. The reactions were started by the addition of 4-nitrophenyl acetate through a multi-channel pipette, and the plate was immediately placed in a CLARIOstar Plus microplate reader (BMG Labtech). Absorbance at 405 nm was recorded every minute for a total duration of 60 min at 25 °C. Data were exported from CLARIOstar Data analysis software (MARS 3.4.1) and further analysed in Prism GraphPad 10.4.1.

## Detection of iridoids by GC-MS
Samples were analysed on an achiral column using an Agilent 8890GC system, an Agilent 5977B GC/MSD (mass selective detector) detector and a CTC Analytics PAL RSI 120 autosampler system. A Zebron ZB-5 Plus column (Phenomenex; internal diameter (ID) = 0.25 mm; length $L$ = 30 m; film thickness = 0.25 µm) was used for separation. Samples (1 µl) were injected at 230 °C inlet temperature. The carrier gas was helium at 1.1 ml min$^{-1}$ constant flow. The temperature gradient was as follows: 2 min at 60 °C, ramp up to 220 °C at 7 °C min$^{-1}$, ramp up to 300 °C at 60 °C min$^{-1}$ and 2 min at 300 °C. The MSD transfer line temperature was 280 °C, and the MS source temperature was 230 °C. After a solvent delay of 6 min, a mass range of 35–250 a.m.u. (atomic mass unit) was collected at 70 eV fragmentation energy.

For confirmation of 7$R$-$cis$-$cis$ nepetalactone identity, samples were analysed on a chiral column using previously established conditions[23]. Briefly, samples were analysed on an Agilent system consisting of an 8890GC system, a 5977B GC/MSD detector, and a 7693 A autosampler. A Supelco β-DEX 225 column (ID = 0.25 mm; $L$ = 30 m; film thickness = 0.25 µm) was used for separation. The samples (1 µl) were injected at 220 °C inlet temperature using a split ratio of 1:10.

Helium was used as carrier gas at a flow rate of 1.1 ml min$^{-1}$. The oven temperature ramp was as follows: 3 min at 80 °C, ramp up to 120 °C at 10 °C min$^{-1}$, 45 min at 120 °C, ramp up to 200 °C at 10 °C min$^{-1}$ and 2 min at 200 °C. The MSD transfer line temperature was 220 °C and the MS source at 230 °C. After 10 min of solvent delay, a mass range of 50–350 a.m.u. was collected at 70 eV fragmentation energy.

GC-MS data were collected using Agilent MassHunter Work Station 10.1.49.

### Synthesis of 7*R-cis-cis* nepetalactone from enzymatically produced 7*R-cis-cis* nepetalactol

(7*R*)-*cis*,*cis*-nepetalactol (7*R*)-*cis*,*cis*-nepetalactone

The combined assays (5 × 100 µl reactions with AmISY and AmICYC performed as described above) were extracted with $CH_2Cl_2$ (100 µl × 3), and the resulting organic phase was then dried over anhydrous $Na_2SO_4$ and partially concentrated under a gentle stream of Ar. Then 4 Å molecular sieve (~5 mg), *N*-methylmorpholine *N*-oxide (1 mg, 8.54 µmol) and tetra-*n*-propylammonium perruthenate (1 mg, 2.85 µmol) were sequentially added to the concentrated organic extract. After stirring under an Ar atmosphere at room temperature for 3 h, the reaction mass was filtered through a short pad of Celite and then directly passed through a short silica column, eluting with $CH_2Cl_2$ (2 ml) and $Et_2O$ (2 ml), to give a colourless eluate that was then partially concentrated under a gentle stream of Ar and directly submitted for chiral GC-MS analysis.

### GC-MS data analysis

Data were analysed in Agilent MassHunter Qualitative Analysis 10.0. Total ion chromatograms were exported as CSV files and imported to Prism Graphpad version 10.4.1 for visualization. Total ion chromatograms shown in the same graph are with the same scaled *y* axis unless otherwise indicated. Peak areas were calculated in Agilent MassHunter Qualitative Analysis 10.0, exported and normalized to internal standard camphor. Bar graphs were constructed in Prism Graphpad version 10.4.1.

### Phylogenetic analyses

Sequences of ICYC orthologues and other MESs were obtained through blast searches against TAIR11, NCBI and One Thousand Plant Transcriptomes (1KP)[91] databases, against transcriptome assemblies from the MintGenomics project[92], or against genome data obtained during this study. Sequences and accession numbers are provided in Supplementary Dataset 1. Full-length amino acid sequences were aligned with webPRANK (https://www.ebi.ac.uk/goldman-srv/webprank/; version updated on 8 October 2017)[93]. Sequence logos were obtained by importing alignments to Geneious Prime 2025.1.2 (Dotmatics). The IQ-TREE webserver (http://iqtree.cibiv.univie.ac.at/) was used to build maximum likelihood phylogenetic trees (automatic substitution model; bootstrap value 1,000) with IQ-Tree 2.3.6 (ref. 94). Trees were visualized in iTOL and graphically edited using itol (https://itol.embl.de/)[95] and Adobe Illustrator 27.8.

### Protein models and Docking

Protein models were predicted by AlphaFold3 through the AlphaFold Server (https://alphafoldserver.com/)[96]. Docking was done using Autodock Vina python version 1.2.5 on the SwissDock webserver (https://www.swissdock.ch/)[97,98]. Models were visualized using ChimeraX version 1.8 for Mac[99].

### Reporting summary

Further information on research design is available in the Nature Portfolio Reporting Summary linked to this article.

### Data availability

All sequencing data associated with this study are available via the NCBI Sequence Read Archive BioProject PRJNA1270996 and PRJNA1169657 (Supplementary Table 1). Reported gene sequences cloned from plant material were deposited to NCBI Genbank under the following accession numbers (Supplementary Table 9): CiGES (PV988048), CiG8H (PV988049), Ci8HGO (PV988050), CiISY (PV988051), Ci7DLGT (PV988052), Ci7DLH (PV988053), CiLAMT (PV988054), CiSLS (PV988055), AsGES (PV988056), AsG8H (PV988057), As8HGO (PV988058), AsISY (PV988059), As7DLGT (PV988060), As7DLH (PV988061), AsSLAS (PV988062), AsICYC (PV988063), CiICYC (PV988064) and CrICYC (PV988065). Source data are provided with this paper.

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

## Acknowledgements

We thank the gardeners E. Rothe and E. Goschala for growing and maintaining *A. salviifolium* and *C. ipecacuanha* plants as well as F. Kaltofen for growing *N. benthamiana* plants. We thank B. Lichman for providing pOPINF_NmMLPL construct, Q. M. Dudley for providing *NmMLPL* and *GPPS* constructs, and P. Sonawane for providing the modified 3Ω1 vector. We are grateful to N. J. Hernández Lozada for providing *7R-cis-cis* nepetalactone standard and C. E. Rodríguez-López for providing *S*-8-oxocitronellal and *R*-8-oxocitronellal. We thank H. Leucke for assistance with RNA extraction, M. Kunert for assistance with UPLC-MS/MS and GC-MS analyses and Y. Nakamura for NMR structural confirmation of commercial *7S-cis-trans* nepetalactol standard. We thank L. Carlton for assistance with nanopore cDNA sequencing. We thank B. Vailllancourt for assistance with sequencing data management. We are grateful to the Max Planck Society for funding and Horizon 2020 (MIAMi, grant number 814645) and the Leibniz Prize, Deutsche Forschungsgemeinschaft (DFG, German Research Foundation) – 505457618 awarded to S.E.O. We acknowledge funding from the University of Georgia, Georgia Research Alliance and Georgia Seed Development to C.R.B.

## Author contributions

M.C. designed all experimental work, analysed data and supervised the work of C.T., C.M., A.D. and J.W. R.P.D. performed RNA extraction, Illumina sequencing genome sequencing and initial genome assemblies. J.P.H. performed genome annotation. M.C. processed tissue-specific gene expression data and performed co-expression analyses. J.C.W. performed snRNA-seq and data processing. K.G. performed VIGS and qPCRs assisted by L.C. R.M.A. performed chemical oxidation of 7-*R-cis-cis* nepetalactol enzymatic product. C.M. performed split-Luciferase assays. C.T., J.W. and A.D. performed cloning. J.W. and A.D. performed recombinant protein purifications. M.C., C.T. and J.W. performed reconstitution in *N. benthamiana*. M.C. performed in vitro enzyme assays. S.H. developed the UPLC-MS/MS method. G.R.T. developed the GC-MS method. A.A.L. provided *C. ipecacuanha* living specimens. M.C., C.R.B. and S.E.O. designed the study. M.C. and S.E.O. wrote the paper with input from all other authors.

## Funding

## Competing interests

The authors declare no competing interests.

## Additional information

**Extended data** is available for this paper at https://doi.org/10.1038/s41477-025-02122-6.

**Correspondence and requests for materials** should be addressed to Maite Colinas, C. Robin Buell or Sarah E. O'Connor.

**a**

Pearson to *CilSY*
expression > 0.8
+
CPM > 50 in young leaves

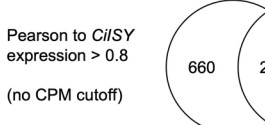

Co-expression module 16
+
> 1 normalized average
expression in cell cluster 30

**List of 13 genes**

| GeneID | functional annotation | name / description | Expression in cell cluster 30 |
|---|---|---|---|
| Caipe.S388690 | alcohol dehydrogenase | Ci8HGO | 4.63 |
| Caipe.S160820 | cytochrome P450, family 76, subfamily C, polypeptide | CiIO | 4.27 |
| **Caipe.S125200** | **methyl esterase** | **CilCYC** | **3.99** |
| Caipe.S125190 | cytochrome P450, family 76, subfamily C, polypeptide | CiG8O | 3.71 |
| Caipe.S184100 | terpene synthase | CiGES | 3.64 |
| Caipe.S366190 | GroES-like zinc-binding dehydrogenase family protein | CYPADH ortholog | 3.45 |
| Caipe.S200000 | cytochrome P450, family 72, subfamily A, polypeptide | Ci7DLH | 3.39 |
| Caipe.S317530 | NAD(P)-binding Rossmann-fold superfamily protein | CilSY paralog | 3.33 |
| Caipe.S244560 | 1-deoxy-D-xylulose 5-phosphate reductoisomerase | DXR ortholog | 3.33 |
| Caipe.S232250 | Deoxyxylulose-5-phosphate synthase | DXS ortholog | 3.26 |
| Caipe.S317540 | NAD(P)-binding Rossmann-fold superfamily protein | CilSY | 3.04 |
| Caipe.S364920 | UDP-glucosyl transferase 85A3 | Ci7DLGT | 2.81 |
| Caipe.S351880 | sulfotransferase | | 2.57 |

**b**

Pearson to *CilSY*
expression > 0.8

(no CPM cutoff)

660 | 22 | 102

Co-expression module 16
+
> 1 normalized average
expression in cell cluster 30

**List of 9 additional genes**

| GeneID | functional annotation | name / description | Expression in cell cluster 30 |
|---|---|---|---|
| Caipe.S317500 | F-BOX WITH WD-40 | | 3.81 |
| Caipe.S135190 | basic helix-loop-helix (bHLH) DNA-binding superfamily protein | BIS ortholog | 2.68 |
| Caipe.S429940 | Phosphatidylinositol-4-phosphate 5-kinase family protein | | 2.37 |
| Caipe.S215190 | Acyl-CoA N-acyltransferases (NAT) superfamily protein | | 2.11 |
| Caipe.S092330 | hypothetical protein | | 2.07 |
| Caipe.S135260 | basic helix-loop-helix (bHLH) DNA-binding superfamily protein | BIS ortholog | 1.82 |
| Caipe.S108310 | basic helix-loop-helix (bHLH) DNA-binding superfamily protein | | 1.42 |
| Caipe.S135200 | basic helix-loop-helix (bHLH) DNA-binding superfamily protein | BIS ortholog | 1.30 |
| Caipe.S352430 | Stress responsive alpha-beta barrel domain protein | | 1.11 |

**Extended Data Fig. 1 | Combination of bulk and single nuclei RNA-seq co-expression analysis.** Because it was unclear which type of enzyme would catalyze this cyclization reaction, we attempted to narrow down the list of candidates through co-expression analyses rather than using functional annotation. To achieve the most highly resolved co-expression list, candidate genes from tissue specific co-expression analysis (Pearson correlation to *CilSY* expression > 0.8) and cell type specific co-expression analysis (co-expression module 16) were overlayed. **a**, In addition to co-expression, only genes with high absolute expression in bulk RNA-seq (>50 CPM in young leaves from plant 1) and snRNA-seq (normalized average expression > 1 in cell cluster 30) were included. The overlay contained the 13 genes listed with functional annotations based on sequence homology. **ICYC** is shown in bold. Blue indicates orthologs of known iridoid pathway genes, green indicates orthologs of known MEP pathway genes. CYPADH has been previously associated with iridoid biosynthesis but its exact function could not be determined (Brown et al, 2015). **b**, Without a cutoff for absolute expression in bulk RNA-seq, the list of genes common to both datasets contained 9 additional genes, including orthologs of the *C. roseus* bHLH iridoid synthesis (BIS) transcription factors that are known to induce expression of IPAP specific iridoid pathway genes[31–33]. Blue indicates orthologs of *C. roseus* BIS. CPM, counts per million;; GES, geraniol synthase; G8H, geraniol 8-hydroxylase; 8HGO, 8-hydroxygeraniol oxidase; ISY, iridoid synthase; ICYC, iridoid cyclase; IO, iridoid oxidase; 7DLGT, 7-deoxyloganetic acid glucosidase; 7DLH, 7-deoxyloganic acid hydroxylase.

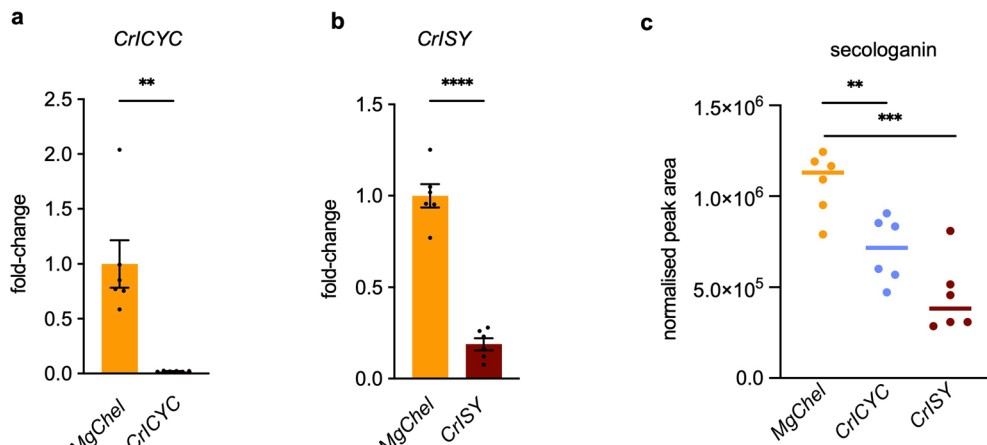

**Extended Data Fig. 2 | Virus Induced Gene Silencing (VIGS) of *ICYC* and *ISY* in *C. roseus*.** Magnesium Chelatase (*MgChel*) was co-silenced in all cases to visualize silenced tissues. In the negative control (in orange) only *MgChel* was silenced. As a positive control *CrISY* was silenced. **a**- **b**, qPCR confirming the downregulation *CrICYC* (**a**) or *CrISY* (**b**) compared to *MgChel* negative control. Expression values are shown as fold-changes relative to expression in *MgChel* negative control. Bar graphs depict mean N = 6 biological replicates, error bars are standard error of the mean. Black dot symbols depict values for individual replicates. P-value of unpaired two-tailed ttest with Welch's test correction: **P = 0.0061, ****P < 0.0001. **c**, Secologanin levels in silenced plants compared to control. LC-MS peak areas normalized to internal standard are shown as individual dots of each replicate, the line depicts the mean of N = 6 biological replicates. P-value of unpaired two-tailed ttest with Welch's test correction: **P = 0.0047, ***P = 0.0002.

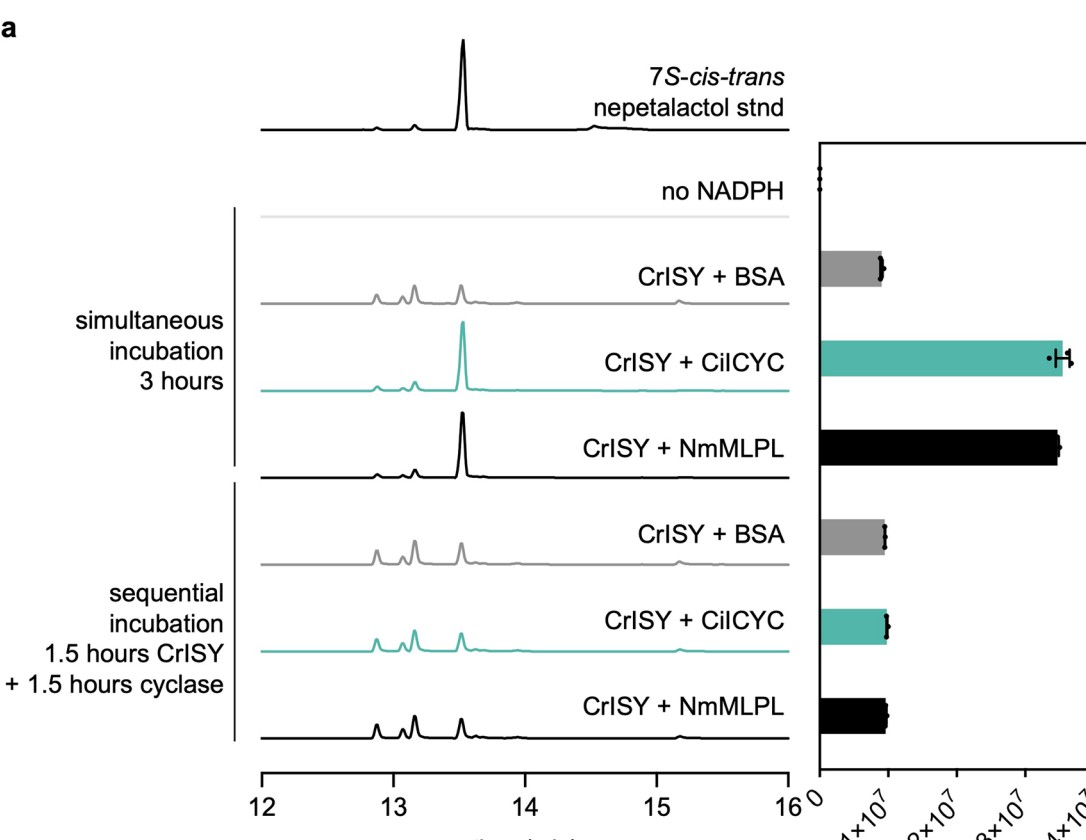

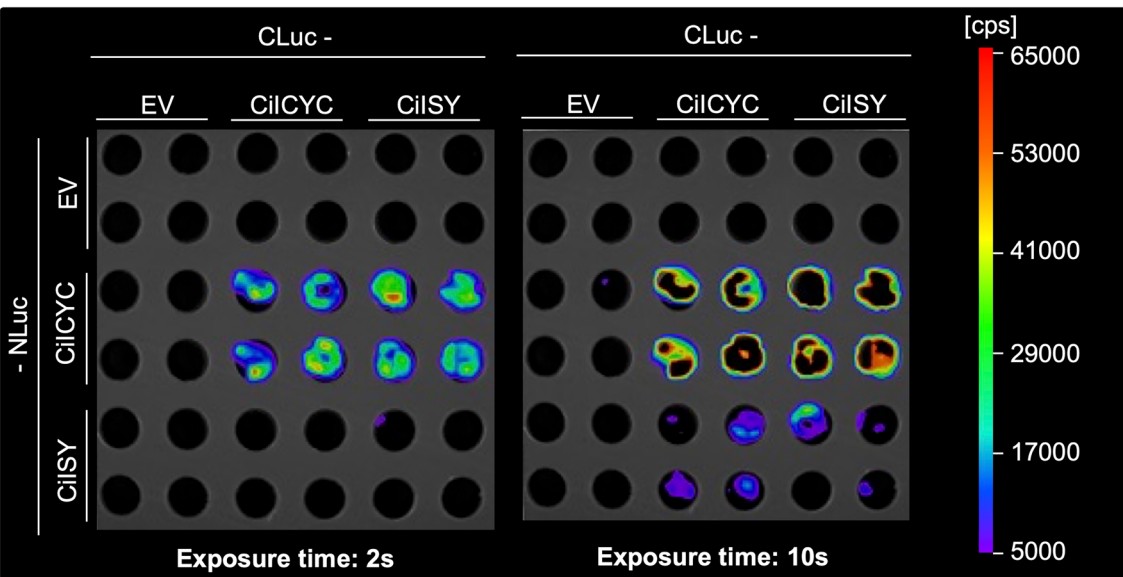

**Extended Data Fig. 3 | ISY and ICYC must be simultaneously present for nepetalactol formation and may interact. a**, Enzyme assays under the conditions shown in Fig. 2a (main text) but with ICYC, NmMLPL or BSA added simultaneously for 3 hours or after 1.5 hours reaction with CrISY alone (sequential incubation). Interestingly, nepetalactol is only formed to higher amounts in simultaneous incubations. This could indicate that the formed ISY product 8-oxocitronellyl enol, which is the substrate for the cyclases, is unstable und must be immediately taken up by the cyclases. Bar graphs depict peak areas from N = 3 replicates of nepetalactol, error bars are standard error of the mean. **b**, Split-Luciferase assay with CiICYC and CiISY in *N. benthamiana*. The C-terminal part of luciferase (CLuc) and the N-terminal part of luciferase (NLuc) are always fused N-terminally or C-terminally, respectively, to the target protein[80]. Empty vectors (EV) contained non-fused CLuc or NLuc, respectively, and served as negative controls. Leaf disks were cut from N = 4 biological replicates of agroinfiltrated *N. benthamiana* plants, transferred to a well plate, where luciferin was added, and imaged using a Nightshade camera (see methods for details). Images are pictures of the same leaf disks with different indicated exposure times. The assays indicate that CiICYC interacts with itself and that CiICYC interacts with CiISY. Note that when NLuc is fused to CiISY interaction is weaker and only revealed at longer exposure times.

**a**

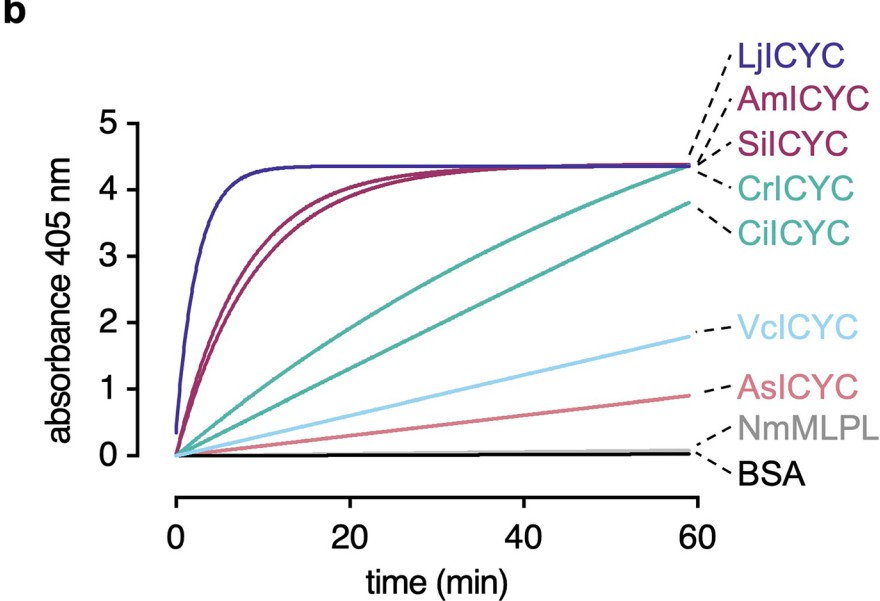

**b**

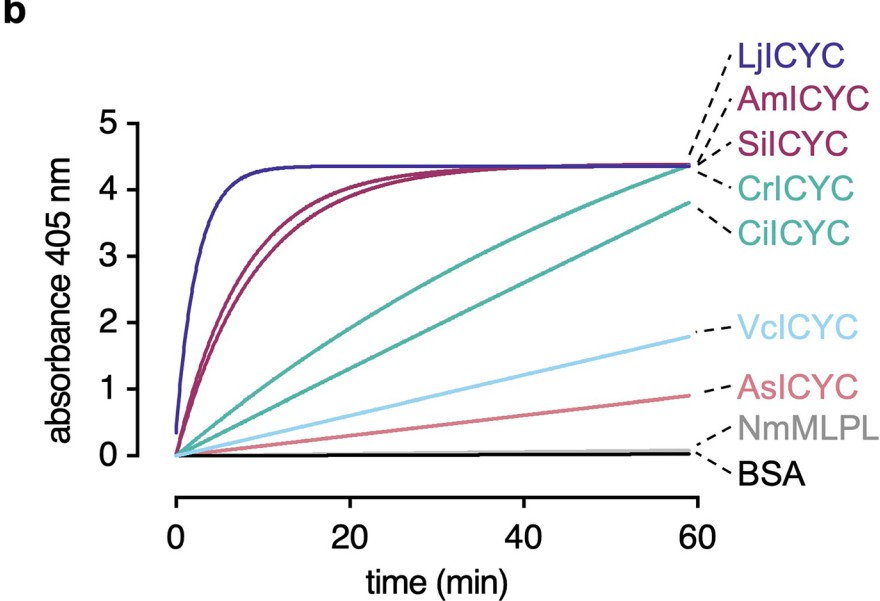

**c**

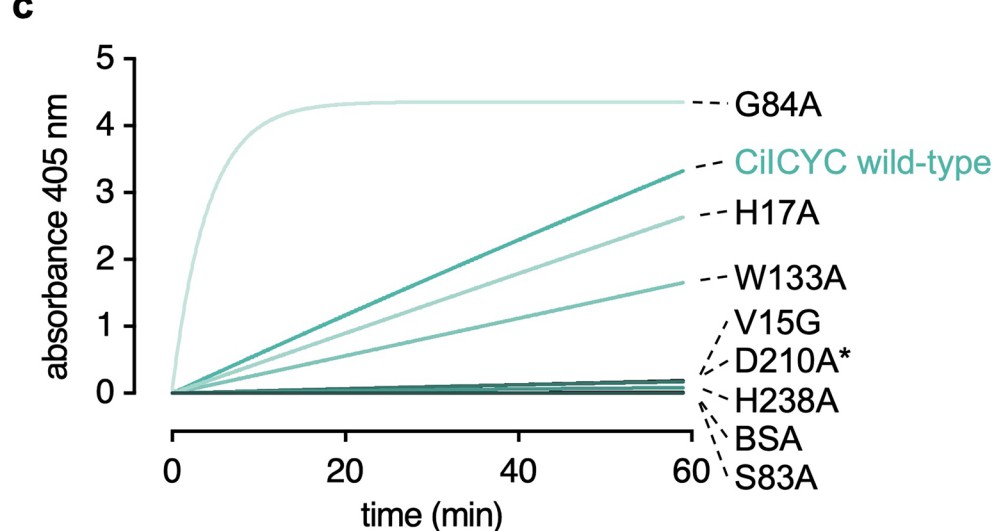

**Extended Data Fig. 4 | See next page for caption.**

**Extended Data Fig. 4 | Esterase activity of ICYC. a**, Reaction scheme of the assay. Esterase activity of the proteins was assessed with 4-nitrophenyl acetate as a substrate. The formation of product is assessed by measuring absorbance of the 4-nitrophenol (yellow) at 405 nm. **b**, Esterase activity assays of 0.5 μM ICYC orthologs compared to NmMLPL and the negative control BSA. Esterase activity greatly varies among ICYC proteins but is detectable for all ICYC orthologs. **c**, Esterase activity assays of 0.5 μM CiICYC wild-type and mutant proteins (see Fig. 3, main text). Mutations in the catalytic triad led to complete abolishment of esterase activity (the asterisk indicates that D210A mutant was poorly soluble, thus activity could not be assessed reliably). Esterase activity appeared to be poorly correlated with cyclase activity (Fig. 3, main text). Each curve was created by curve fitting of absorbance values of N = 3 replicates. Absorbance was measured every minute on a microplate reader (see methods for details).

C. Robin Buell
Sarah E. O'Connor

# Reporting Summary

## Statistics

For all statistical analyses, confirm that the following items are present in the figure legend, table legend, main text, or Methods section.

| n/a | Confirmed | |
|---|---|---|
| ☐ | ☒ | The exact sample size (*n*) for each experimental group/condition, given as a discrete number and unit of measurement |
| ☐ | ☒ | A statement on whether measurements were taken from distinct samples or whether the same sample was measured repeatedly |
| ☐ | ☒ | The statistical test(s) used AND whether they are one- or two-sided *Only common tests should be described solely by name; describe more complex techniques in the Methods section.* |
| ☒ | ☐ | A description of all covariates tested |
| ☒ | ☐ | A description of any assumptions or corrections, such as tests of normality and adjustment for multiple comparisons |
| ☐ | ☒ | A full description of the statistical parameters including central tendency (e.g. means) or other basic estimates (e.g. regression coefficient) AND variation (e.g. standard deviation) or associated estimates of uncertainty (e.g. confidence intervals) |
| ☐ | ☒ | For null hypothesis testing, the test statistic (e.g. *F*, *t*, *r*) with confidence intervals, effect sizes, degrees of freedom and *P* value noted *Give P values as exact values whenever suitable.* |
| ☒ | ☐ | For Bayesian analysis, information on the choice of priors and Markov chain Monte Carlo settings |
| ☒ | ☐ | For hierarchical and complex designs, identification of the appropriate level for tests and full reporting of outcomes |
| ☐ | ☒ | Estimates of effect sizes (e.g. Cohen's *d*, Pearson's *r*), indicating how they were calculated |

*Our web collection on statistics for biologists contains articles on many of the points above.*

## Software and code

Policy information about availability of computer code

| Data collection | All presented data have been acquired using existing and routinely used software. LC-MS data was collected with Bruker Compass qtofControl 5.2.109 / Hystar 5.1.5.1 or Compass qtofControl 6.3 / Hystar 6.0.30.0 software. GC-MS data was collected using Agilent MassHunter Work Station 10.1.49. Short read RNA-seq and DNA-seq data and single nuclei RNA-seq were sequenced on an Illumina NovaSeq 6000 PE150 platform. Long read DNA-seq was sequenced on a on an R10.4.1 PromethION flowcell (FLO-PRO114M; Oxford Nanopore Technologies (ONT), Oxford, UK). Long read cDNA was sequenced on a MIN106 Rev. D flowcell (Oxford Nanopore Technologies (ONT), Oxford, UK). Split Luciferase data was collected with indiGOTM 1.4 software (Berthold Technologies). |
|---|---|

| Data analysis | All data analysis was done using routinely used software. For phylogenetic trees sequences were aligned with webPRANK (https://www.ebi.ac.uk/goldman-srv/webprank/; version updated 8 Oct 2017). Phylogenetic trees were constructed using IQ-Tree 2.3.6 (http://iqtree.cibiv.univie.ac.at/). Protein structural models were predicted using AlphaFold 3 (https://alphafoldserver.com). Docking was done using AudoDock Vina python version 1.2.5 on the SwissDock webserver (https://www.swissdock.ch). Structures were visualized using ChimeraX 1.8. LC-MS data was analysed using MZmine 4.5.37. GC-MS data was analyzed using Agilent MassHunter Qualitative Analysis 10.0. Chemical structures were generated in ChemDraw Professional 23.1.12. Short read RNA-seq data was quality checked with FastQC (Galaxy Version 0.73), quality trimmed with Trimmomatic (Galaxy Version 0.38.1). Oxford Nanopore (ONT) cDNA reads were processed with Pychopper v2.7.10. ONT genomic reads were assembled using Flye v.2.9.1. For genome annotation genome assembly data was processed and analyzed using RepeatModeler 2.03, ProtExcuder 1.2, Cutadapt 2.10, HISAT2 2.1.0, minimap2 2.17-r941, Stringtie 2.2.1, BRAKER2 2.1.6, PASA2 2.5.2. Functional annotation of transcripts was performed using BLASTP 2.12.0 and PfamScan 1.6. RNA-seq expression values were generated in CLC Genomics Workbench 24.0.1 (Qiagen). Single nuclei transcriptomics data was processed with pipseeker 3.1.3, STARsolo 2.7.10b, Seurat 4.3.0.1. Co-expression analysis was done with Simple Tidy_GeneCoEx. All graphs and heatmaps were prepared with GraphPad Prism 10.4.1 or 10.4.2. Main figures were assembled in Adobe Illustrator 27.8. Supplementary figures were assembled in PowerPoint 16.89.1. |
|---|---|

For manuscripts utilizing custom algorithms or software that are central to the research but not yet described in published literature, software must be made available to editors and reviewers. We strongly encourage code deposition in a community repository (e.g. GitHub). See the Nature Portfolio guidelines for submitting code & software for further information.

## Data

Policy information about availability of data

All manuscripts must include a data availability statement. This statement should provide the following information, where applicable:
- Accession codes, unique identifiers, or web links for publicly available datasets
- A description of any restrictions on data availability
- For clinical datasets or third party data, please ensure that the statement adheres to our policy

All sequencing data associated with this study are available at the National Center for Biotechnology Information (NCBI) Sequence Read Archive BioProject PRJNA1270996 and PRIJNA1169657 (Supplementary Table 1). Reported gene sequences cloned from plant material were deposited to NCBI Genbank under the following accession numbers (Supplementary Table 9): CiGES (PV988048), CiG8H (PV988049), Ci8HGO (PV988050), CiISY (PV988051), Ci7DLGT (PV988052), Ci7DLH (PV988053), CiLAMT (PV988054), CiSLS (PV988055), AsGES (PV988056), AsG8H (PV988057), As8HGO (PV988058), AsISY (PV988059), As7DLGT (PV988060), As7DLH (PV988061), AsSLAS (PV988062), AsICYC (PV988063), CiICYC (PV988064), and CrICYC (PV988065).

## Research involving human participants, their data, or biological material

Policy information about studies with human participants or human data. See also policy information about sex, gender (identity/presentation), and sexual orientation and race, ethnicity and racism.

| Reporting on sex and gender | Not applicable |
|---|---|
| Reporting on race, ethnicity, or other socially relevant groupings | Not applicable |
| Population characteristics | Not applicable |
| Recruitment | Not applicable |
| Ethics oversight | Not applicable |

Note that full information on the approval of the study protocol must also be provided in the manuscript.

## Field-specific reporting

Please select the one below that is the best fit for your research. If you are not sure, read the appropriate sections before making your selection.

☒ Life sciences  ☐ Behavioural & social sciences  ☐ Ecological, evolutionary & environmental sciences

For a reference copy of the document with all sections, see nature.com/documents/nr-reporting-summary-flat.pdf

## Life sciences study design

All studies must disclose on these points even when the disclosure is negative.

| Sample size | Prior determination of sample size is not applicable to this study. Nicotiana benthamiana pathway reconstitution experiments were done on three independent biological replicates which is standard for this type of experiments. In vitro assays were performed as three technical replicates from the same enzyme purifications which is standard for this type of experiments. Bulk RNA-seq was performed on one replicate which is sufficient for gene discovery as no statistics were applied. Single-nuclei RNA-seq was performed on two biological replicates (i.e. individual plants) of Carapichea ipecacuanha which is standard for this type of analysis. |
|---|---|
| Data exclusions | No data was excluded from the analyses. |

| Replication | Details about biological replicates are provided in the figure legends. Pathway reconstitution experiments were conducted on three biological replicates that correspond to three independent individual plants. All attempts of replication were successful. In vitro assays were performed as three technical replicates from the same enzyme purifications. Replications with different enzyme purifications were successful. |
|---|---|
| Randomization | The order of all LC-MS samples was randomized prior to the runs. Plants were grown in randomized order. Experiments were successfully repeated on other days. For other types of experiments of this study randomization is not relevant. |
| Blinding | Blinding was not relevant for this study; characterization of pathway genes and enzymes requires insight into the experimental conditions and characteristics of the samples. |

# Reporting for specific materials, systems and methods

We require information from authors about some types of materials, experimental systems and methods used in many studies. Here, indicate whether each material, system or method listed is relevant to your study. If you are not sure if a list item applies to your research, read the appropriate section before selecting a response.

## Materials & experimental systems

| n/a | Involved in the study |
|---|---|
| ☒ | Antibodies |
| ☒ | Eukaryotic cell lines |
| ☒ | Palaeontology and archaeology |
| ☒ | Animals and other organisms |
| ☒ | Clinical data |
| ☒ | Dual use research of concern |
| ☐ | ☒ Plants |

## Methods

| n/a | Involved in the study |
|---|---|
| ☒ | ChIP-seq |
| ☒ | Flow cytometry |
| ☒ | MRI-based neuroimaging |

## Dual use research of concern

Policy information about dual use research of concern

### Hazards

Could the accidental, deliberate or reckless misuse of agents or technologies generated in the work, or the application of information presented in the manuscript, pose a threat to:

| No | Yes | |
|---|---|---|
| ☒ | ☐ | Public health |
| ☒ | ☐ | National security |
| ☒ | ☐ | Crops and/or livestock |
| ☒ | ☐ | Ecosystems |
| ☒ | ☐ | Any other significant area |

### Experiments of concern

Does the work involve any of these experiments of concern:

| No | Yes | |
|---|---|---|
| ☒ | ☐ | Demonstrate how to render a vaccine ineffective |
| ☒ | ☐ | Confer resistance to therapeutically useful antibiotics or antiviral agents |
| ☒ | ☐ | Enhance the virulence of a pathogen or render a nonpathogen virulent |
| ☒ | ☐ | Increase transmissibility of a pathogen |
| ☒ | ☐ | Alter the host range of a pathogen |
| ☒ | ☐ | Enable evasion of diagnostic/detection modalities |
| ☒ | ☐ | Enable the weaponization of a biological agent or toxin |
| ☒ | ☐ | Any other potentially harmful combination of experiments and agents |

# Plants

| | |
|---|---|
| Seed stocks | Nicotiana benthamiana and Catharanthus roseus seeds were obtained from seed stocks maintained by the greenhouse team at Max Planck Institute for Chemical Ecology, Jena. Carapichea ipecacuanha and Alangium salviifolium were obtained as plantlets, no seed stocks were obtained. No other plant species were grown for this study. |
| Novel plant genotypes | No stable transformation was carried out. Heterologous overexpression in N. benthamiana was done transiently through leaf agroinfiltration as described in the methods paragraph "A. tumefaciens mediated transient expression in N. benthamiana". Virus-induced gene silencing in Catharanthus roseus was also transient as described in the methods paragraph "Virus induced gene silencing (VIGS) in C. roseus". |
| Authentication | Transient transformation of N. benthamiana through leaf agroinfiltration was done as described in the methods paragraph "A. tumefaciens mediated transient expression in N. benthamiana". Virus-induced gene silencing in Catharanthus roseus was also transient as described in the methods paragraph "Virus induced gene silencing (VIGS) in C. roseus". No stable transformation was carried out. |

