## [Peer Review File · Nature Plants]

Discovery of iridoid cyclase completes the iridoid pathway in asterids

Corresponding Author: Professor Sarah O'Connor

Version 0:

Decision Letter:

21st May 2025

Dear Professor O'Connor,

Thank you very much for your enquiry about submitting your manuscript "Discovery of iridoid cyclase completes the iridoid pathway in asterids" to Nature Plants. It certainly sounds interesting, and we would be happy to consider it for publication. However, I'm sure you'll understand that we cannot make a firm decision about whether to send the paper out to review until we have carefully read the full paper (and appropriate background literature).

In order to submit your complete manuscript to Nature Plants, please use the link below:

Link Redacted

If you have any questions, please feel free to contact me.

Yours sincerely,

Version 1:

Reviewer comments:

Reviewer #1

(Comments for the Author)

The manuscript entitled "Discovery of iridoid cyclase completes the iridoid pathway in asterids" (NPLANTS-250519826A) by Colinas et al. that has been submitted for consideration in the Nature Plants, presents data on the discovery of novel enzymes in asterids - iridoid cyclases (ICYC), which catalyze cyclization of the reactive 3S- or 3R-8-oxocitronellyl enol to form 7S-cis-trans and 7R-cis-cis-nepetalactol. It has been known that this cyclisation occurs spontaneously at low yields, and that in *Nepeta* species it is catalyzed by NEPS cyclases and MLPLs, however no enzymes with this function have previously been characterized in other iridoid-producing plants. According to the authors, the ICYCs belong to the methyl esterase (MES) type / hydrolases, which are entirely unrelated to *Nepeta*-specific cyclases, and are phylogenetically well distinguished from other methylesterases. However, they contain functional catalytic triad (serine, aspartate/glutamate and histidine) and a glycine-rich oxyanion hole motifs which are characteristic for canonical esterases, and are responsible for the catalysis of substrate hydrolysis. For the first time, Colinas et al. associate these group of enzymes with the cyclization function, and provide unambiguous proofs for their catalytic activity using in vitro enzymatic assays of the mutant proteins, reconstitution of the secoiridoid pathway in *C. ipecacuanha* and *N. benthamiana*, VIGS silencing of CrICYCs in native plant - *C. roseus*, and the molecular docking analysis. Adopting the split luciferase assays, the authors further prove the interaction of ISY and ICYC, thus suggesting the substrate channeling. The results are well presented and discussed, the methodology is appropriate, and statistical data analysis is adequate.

Although the mechanistic basis of the cyclization mediated by ICYCs needs further explanations and experimental proofs, the contents of the manuscript in the present form are original, and will be of interest to a wider community of researchers. The major findings of the Manuscript are novel, and they represent a significant advance in the field, as they fulfil the knowledge gaps about the important steps of the early iridoid biosynthesis pathway in asterids. The results also have applicative

potential, as they open a vast array of possibilities for metabolic engineering of plants towards the production of different stereoisomers of iridoids.

I strongly recommend this manuscript for publication in Nature Plants, pending minor revisions:

-Abstract: The overall impression is that the abstract relies too heavily on previously published literature and does not sufficiently emphasize the novel findings of the current study. I strongly recommend that the authors revise the abstract to clearly highlight their main discoveries, the significance of their results, and the study's broader contribution to our understanding of iridoid biosynthesis. In particular, please elaborate on how the knowledge generated in this work could facilitate future efforts to uncover and understand the diversity of iridoids across the plant kingdom, and thus their chemical evolution. Be more precise in explaining which reactive intermediates are cyclized by ICYCs, and what is the stereochemistry of resulting iridoids (Lines 33-34). I also suggest to authors to change "iridoid biosynthesis pathway in asterides" into "early iridoid biosynthesis pathway in asterides" (Line 35).

Main: The conclusion paragraph could be improved.

Reviewer #2

(Comments for the Author)

The authors reported the identification and functional characterization of ICYCs, which fill the biosynthetic gap between ISY- and IO-catalyzed reaction steps involved in the biosynthesis of iridoid in asterid plants. The results are interesting for biosynthetic pathway elucidation of plant-derived compounds and the experimentals are well-designed and performed. There are some problems should be clarified.

Major:

1) ISY catalyzes the 1,4-reduction of 8-oxo-generial to form 8-oxo-citronellyl enol. The latter compound is not stable in vitro and a spontaneous cyclization will be taken place, leading to the formation of nepetalactol. However, two kinds of enzymes, MLPL and NEPS, had been reported to catalyze the cyclization of 8-oxo-citronellyl enol to give nepetalactol. NEPS will catalyze the proceeding dehydrogenation of nepetalactol to give nepetalactone as the end product. The authors had clarified that ICYC is not related to NEPS in the manuscript. How about MLPL? Additionally, the novelty of the ICYC-catalyzed reaction is limited since NEPS and MLPL can catalyze the cyclization of 8-oxo-citronellyl enol.

2) Fig 1e and 3d, there is not any loganic acid was detected by LC-MS from the combination reaction without ICYC. The result is not consistent with that a small amount of nepetalactol produced from the spontaneous cyclization of 8-oxo-citronellyl enol, which will offer a small amount of loganic acid, together with the following enzymes such as IO, 7DLGT, 7DLH.

Minor:

1) Fig 2d, an oxygen atom was omitted in the chemical structure of 7R-cis-cis nepetalactone.

2) L101-102, L128-129, it is not necessary to capitalize the full name of 7DLH and 7DLGT.

3) L145, the CiIYC should be CiIICYC.

4) L276-279, the sentence should be revised.

5) For the information for references, the authors should check them carefully.

Reviewer #3

(Comments for the Author)

The manuscript by Colinas and colleagues reports the discovery of iridoid cyclases (ICYC) that catalyze an unusual and hitherto hidden reaction in the biosynthesis of iridoid natural products in asterids. This work illuminates the cyclization of the reactive enol intermediate as a key step in iridoid formation, thus completing the known iridoid biosynthetic pathway. Given the natural functions and medicinal importance of iridoids, this study is timely and should be of keen interest to a broad audience. The presented conclusions are overall well supported, and the experiments have been elegantly performed. The manuscript is very well written and illustrated.

Below are number of commenst that the authors might find helpful to further strengthen the manuscript.

- Line 42: Consider removing 'widespread', since iridoids are largely restricted to asterid species.

- Line 45: Consider adding long-distance vs. local herbivore defenses.

- Line 116: Consider specifically mentioning DXS and DXR here, highlighting that the 13 genes include major rate-limiting enzymes.

- Line 155: Were other metabolic alterations observed that may suggest that accumulating intermediates are metabolized through compatible pathways?

- Line 158: Supplementary Fig. 10 provides some important insights and might be valuable to be included as a main figure.

- Fig. 1: For part (d) it is a bit hard to see how the genes align with the diagram. For part (e) please include a unit for the y axis.

- Line 218: While split luciferase assays should suffice to demonstrate protein interactions in the context of this study, they do not provide evidence for substrate channeling. I would recommend commenting on this more clearly in the discussion. If protein structural and/or subcellular localization data are available, these would provide valuable additional support for this hypothesis.

- Line 261: Is the esterase activity also relevant in planta? For example, did metabolite profiling of VIGS plants indicate alterations in any other than the focal iridoid pathway?

Decision Letter:

18th July 2025

Dear Professor O'Connor,

Your Letter, "Discovery of iridoid cyclase completes the iridoid pathway in asterids" has now been seen by three referees. You will see from their comments below that while they find your work of interest, some important points are raised. We are interested in the possibility of publishing your study in Nature Plants, but would like to consider your response to these concerns in the form of a revised manuscript before we make a final decision on publication.

While reviewers #1 and #3 have predominantly minor concerns and suggestions, reviewer #2 raises some issues that require further clarification. We therefore invite you to revise your manuscript taking into account all reviewer and editor comments. Please highlight all changes in the manuscript text file, preferably in Microsoft Word format.

When revising your manuscript please:

* If you have not done so already please begin to revise your manuscript so that it conforms to our Letter format instructions at <http://www.nature.com/nplants/info/final-submission>. Refer also to any guidelines provided in this letter.

* Pay close attention to our [href="https://www.nature.com/nature-portfolio/editorial-policies/image-integrity">Digital Image Integrity Guidelines](https://www.nature.com/nature-portfolio/editorial-policies/image-integrity) and to the following points. Please ensure:

-- that unprocessed scans are clearly labelled and match the gels and western blots presented in figures.

-- that control panels for gels and western blots are appropriately described as loading or sample processing controls

-- that all images in the paper are checked for duplication of panels and for splicing of gel lanes.

-- that you retain unprocessed data and metadata files after publication, ideally archiving data in perpetuity. These may be requested during the peer review and production process or after publication if any issues arise.

EXTENDED DATA FIGURES

Nature Plants strongly supports public availability of data and are therefore keen that the data used in your paper is placed in an appropriate public data repository. Alternatively, if this is not possible, you may present the data as Extended Data or Supplementary Information. If data can only be shared on request, please explain why in your Data Availability Statement, and also in the correspondence with your editor. Please note that for some data types, deposition in a public repository is mandatory.

Link Redacted

We hope to receive your revised manuscript within four to eight weeks. If you cannot send it within this time, please let us know. We will be happy to consider your revision so long as nothing similar has been accepted for publication at Nature Plants or published elsewhere.

Sincerely,

Nature Plants is committed to improving transparency in authorship. As part of our efforts in this direction, we are now requesting that all authors identified as 'corresponding author' on published papers create and link their Open Researcher and Contributor Identifier (ORCID) with their account on the Manuscript Tracking System (MTS), prior to acceptance. This applies to primary research papers only. ORCID helps the scientific community achieve unambiguous attribution of all scholarly contributions. You can create and link your ORCID from the home page of the MTS by clicking on 'Modify my Springer Nature account'. For more information please visit www.springernature.com/orcid.

Reviewers' Comments:

Reviewer #1 (Comments for the Author):

The manuscript entitled "Discovery of iridoid cyclase completes the iridoid pathway in asterids" (NPLANTS-250519826A) by Colinas et al. that has been submitted for consideration in the Nature Plants, presents data on the discovery of novel enzymes in asterids - iridoid cyclases (ICYC), which catalyze cyclization of the reactive 3S- or 3R-8-oxocitronellyl enol to form 7S-cis-trans and 7R-cis-cis-nepetalactol. It has been known that this cyclisation occurs spontaneously at low yields, and that in *Nepeta* species it is catalyzed by NEPS cyclases and MLPLs, however no enzymes with this function have previously been characterized in other iridoid-producing plants. According to the authors, the ICYCs belong to the methyl esterase (MES) type / hydrolases, which are entirely unrelated to *Nepeta*-specific cyclases, and are phylogenetically well distinguished from other methyl esterases. However, they contain functional catalytic triad (serine, aspartate/glutamate and histidine) and a glycine-rich oxyanion hole motifs which are characteristic for canonical esterases, and are responsible for the catalysis of substrate hydrolysis. For the first time, Colinas et al. associate these group of enzymes with the cyclization function, and provide unambiguous proofs for their catalytic activity using in vitro enzymatic assays of the mutant proteins, reconstitution of the secoiridoid pathway in *C. ipecacuanha* and *N. benthamiana*, VIGS silencing of CrICYCs in native plant - *C. roseus*, and the molecular docking analysis. Adopting the split luciferase assays, the authors further prove the interaction of ISY and ICYC, thus suggesting the substrate channeling. The results are well presented and discussed, the methodology is appropriate, and statistical data analysis is adequate.

Although the mechanistic basis of the cyclization mediated by ICYCs needs further explanations and experimental proofs, the contents of the manuscript in the present form are original, and will be of interest to a wider community of researchers. The major findings of the Manuscript are novel, and they represent a significant advance in the field, as they fulfil the knowledge gaps about the important steps of the early iridoid biosynthesis pathway in asterids. The results also have applicative potential, as they open a vast array of possibilities for metabolic engineering of plants towards the production of different stereoisomers of iridoids.

I strongly recommend this manuscript for publication in Nature Plants, pending minor revisions:

-Abstract: The overall impression is that the abstract relies too heavily on previously published literature and does not sufficiently emphasize the novel findings of the current study. I strongly recommend that the authors revise the abstract to clearly highlight their main discoveries, the significance of their results, and the study's broader contribution to our understanding of iridoid biosynthesis. In particular, please elaborate on how the knowledge generated in this work could facilitate future efforts to uncover and understand the diversity of iridoids across the plant kingdom, and thus their chemical evolution. Be more precise in explaining which reactive intermediates are cyclized by ICYCs, and what is the stereochemistry of resulting iridoids (Lines 33-34). I also suggest to authors to change "iridoid biosynthesis pathway in asterides" into "early iridoid biosynthesis pathway in asterides" (Line 35).

Main: The conclusion paragraph could be improved.

Reviewer #2 (Comments for the Author):

The authors reported the identification and functional characterization of ICYCs, which fill the biosynthetic gap between ISY- and IO-catalyzed reaction steps involved in the biosynthesis of iridoid in asterid plants. The results are interesting for

biosynthetic pathway elucidation of plant-derived compounds and the experimentals are well-designed and performed. There are some problems should be clarified.

Major:

1) ISY catalyzes the 1,4-reduction of 8-oxo-generial to form 8-oxo-citronellyl enol. The latter compound is not stable in vitro and a spontaneous cyclization will be taken place, leading to the formation of nepetalactol. However, two kinds of enzymes, MLPL and NEPS, had been reported to catalyze the cyclization of 8-oxo-citronellyl enol to give nepetalactol. NEPS will catalyze the proceeding dehydrogenation of nepetalactol to give nepetalactone as the end product. The authors had clarified that ICYC is not related to NEPS in the manuscript. How about MLPL? Additionally, the novelty of the ICYC-catalyzed reaction is limited since NEPS and MLPL can catalyze the cyclization of 8-oxo-citronellyl enol.

2) Fig 1e and 3d, there is not any loganic acid was detected by LC-MS from the combination reaction without ICYC. The result is not consistent with that a small amount of nepetalactol produced from the spontaneous cyclization of 8-oxo-citronellyl enol, which will offer a small amount of loganic acid, together with the following enzymes such as IO, 7DLGT, 7DLH.

Minor:

1) Fig 2d, an oxygen atom was omitted in the chemical structure of 7R-cis-cis nepetalactone.

2) L101-102, L128-129, it is not necessary to capitalize the full name of 7DLH and 7DLGT.

3) L145, the CiIYC should be CiIICYC.

4) L276-279, the sentence should be revised.

5) For the information for references, the authors should check them carefully.

Reviewer #3 (Comments for the Author):

The manuscript by Colinas and colleagues reports the discovery of iridoid cyclases (ICYC) that catalyze an unusual and hitherto hidden reaction in the biosynthesis of iridoid natural products in asterids. This work illuminates the cyclization of the reactive enol intermediate as a key step in iridoid formation, thus completing the known iridoid biosynthetic pathway. Given the natural functions and medicinal importance of iridoids, this study is timely and should be of keen interest to a broad audience. The presented conclusions are overall well supported, and the experiments have been elegantly performed. The manuscript is very well written and illustrated.

Below are number of commenst that the authors might find helpful to further strengthen the manuscript.

- Line 42: Consider removing 'widespread', since iridoids are largely restricted to asterid species.

- Line 45: Consider adding long-distance vs. local herbivore defenses.

- Line 116: Consider specifically mentioning DXS and DXR here, highlighting that the 13 genes include major rate-limiting enzymes.

- Line 155: Were other metabolic alterations observed that may suggest that accumulating intermediates are metabolized through compatible pathways?

- Line 158: Supplementary Fig. 10 provides some important insights and might be valuable to be included as a main figure.

- Fig. 1: For part (d) it is a bit hard to see how the genes align with the diagram. For part (e) please include a unit for the y axis.

- Line 218: While split luciferase assays should suffice to demonstrate protein interactions in the context of this study, they do not provide evidence for substrate channeling. I would recommend commenting on this more clearly in the discussion. If protein structural and/or subcellular localization data are available, these would provide valuable additional support for this hypothesis.

- Line 261: Is the esterase activity also relevant in planta? For example, did metabolite profiling of VIGS plants indicate alterations in any other than the focal iridoid pathway?

*****END*****

Version 2:

Reviewer comments:

Reviewer #1

(Comments for the Author)

Dear Editor,

I have reviewed the revised manuscript titled "Discovery of iridoid cyclase completes the iridoid pathway in asterids" (NPLANTS-250519826A) by Colinas et al. The authors have satisfactorily addressed the concerns raised in the initial round of review through their responses and revisions. In my opinion, the manuscript is now suitable for publication in Nature Plants as a Letter.

Reviewer #2

(Comments for the Author)

The authors addressed the questions raised from the first-round review. Here are some minor problems need to fix. For e.g., L43, the sentence should be re-written since there are two "widespread". L52, geraniol-diphosphate should be geranyl diphosphate. L82-83, the sentence should be re-written... The authors should double check the English context of the manuscript.

Reviewer #3

(Comments for the Author)

I appreciate the clarifications and additional data provided in this revised manuscript. I believe these have strengthened the study and satisfied my comments on the original submission. I think this manuscript will make an important contribution to the field.

Decision Letter:

Our ref: NPLANTS-250519826B

12th August 2025

Dear Dr. O'Connor,

Thank you for submitting your revised manuscript "Discovery of iridoid cyclase completes the iridoid pathway in asterids" (NPLANTS-250519826B). It has now been seen by the original referees and their comments are below. The reviewers find that the paper has improved in revision, and therefore we'll be happy in principle to publish it in Nature Plants, pending minor revisions to satisfy the referees' final requests and to comply with our editorial and formatting guidelines.

Thank you again for your interest in Nature Plants Please do not hesitate to contact me if you have any questions.

Sincerely,

Reviewer #1 (Comments for the Author):

I have reviewed the revised manuscript titled "Discovery of iridoid cyclase completes the iridoid pathway in asterids" (NPLANTS-250519826A) by Colinas et al. The authors have satisfactorily addressed the concerns raised in the initial round of review through their responses and revisions. In my opinion, the manuscript is now suitable for publication in Nature Plants as a Letter.

Reviewer #2 (Comments for the Author):

The authors addressed the questions raised from the first-round review. Here are some minor problems need to fix. For e.g., L43, the sentence should be re-written since there are two "widespread". L52, geraniol-diphosphate should be geranyl diphosphate. L82-83, the sentence should be re-written... The authors should double check the English context of the

manuscript.

Reviewer #3 (Comments for the Author):

I appreciate the clarifications and additional data provided in this revised manuscript. I believe these have strengthened the study and satisfied my comments on the original submission. I think this manuscript will make an important contribution to the field.

Version 3:

Decision Letter:

3rd September 2025

Dear Professor O'Connor,

We are pleased to inform you that your Letter entitled "Discovery of iridoid cyclase completes the iridoid pathway in asterids", has now been accepted for publication in Nature Plants.

Over the next few weeks, your paper will be copyedited to ensure that it conforms to Nature Plants style. We look particularly carefully at the titles of all papers to ensure that they are relatively brief and understandable.

Once your paper is typeset, you will receive an email with a link to choose the appropriate publishing options for your paper and our Author Services team will be in touch regarding any additional information that may be required.

Acceptance of your manuscript is conditional on all authors' agreement with our publication policies (see <http://www.nature.com/authors/policies/index.html>). In particular your manuscript must not be published elsewhere.

Authors may need to take specific actions to achieve compliance with funder and institutional open access

mandates. If your research is supported by a funder that requires immediate open access (e.g. according to [a href="https://www.springernature.com/gp/open-science/plan-s-compliance">Plan S principles](https://www.springernature.com/gp/open-science/plan-s-compliance) or the [a href="https://www.springernature.com/gp/open-science/us-federal-agency-compliance">NIH public access policy](https://www.springernature.com/gp/open-science/us-federal-agency-compliance)) then you should select the gold OA route, and we will direct you to the compliant route where possible. Because authors warrant under our subscription licensing terms that they haven't committed to licensing any version of their article under a licence inconsistent with the terms of our agreement – including the applicable embargo period – publication under the subscription model isn't suitable for authors whose funders require no embargo.

An online order form for reprints of your paper is available at [a href="https://www.nature.com/reprints/author-reprints.html">https://www.nature.com/reprints/author-reprints.html](https://www.nature.com/reprints/author-reprints.html). All co-authors, authors' institutions and authors' funding agencies can order reprints using the form appropriate to their geographical region.

We welcome the submission of potential cover material (including a short caption of around 40 words) related to your manuscript; suggestions should be sent to Nature Plants as electronic files (the image should be 300 dpi at 210 x 297 mm in either TIFF or JPEG format). Please note that such pictures should be selected more for their aesthetic appeal than for their

scientific content, and that colour images work better than black and white or grayscale images. Please include a written description of your image and how it was created along with your image files. Please do not try to design a cover with the Nature Plants logo etc., and please do not submit composites of images related to your work. I am sure you will understand that we cannot make any promise as to whether any of your suggestions might be selected for the cover of the journal.

With kind regards,

P.S. Click on the following link if you would like to recommend Nature Plants to your librarian
<http://www.nature.com/subscriptions/recommend.html#forms>

** Visit the Springer Nature Editorial and Publishing website at http://editorial-jobs.springernature.com?utm_source=ejP_NPlan_email&utm_medium=ejP_NPlan_email&utm_campaign=ejp_NPlan for more information about our career opportunities. If you have any questions please click [here](mailto:editorial.publishing.jobs@springernature.com).**

NPLANTS-250519826A – Response to referees

We thank the three reviewers for their thoughtful and constructive comments and the time it took to carefully review this manuscript. Below is a point-by-point response to their comments.

Reviewer #1:

*The manuscript entitled “Discovery of iridoid cyclase completes the iridoid pathway in asterids” (NPLANTS-250519826A) by Colinas et al. that has been submitted for consideration in the Nature Plants, presents data on the discovery of novel enzymes in asterids - iridoid cyclases (ICYC), which catalyze cyclization of the reactive 3S- or 3R-8-oxocitronellyl enol to form 7S-cis-trans and 7R-cis-cis-nepetalactol. It has been known that this cyclisation occurs spontaneously at low yields, and that in Nepeta species it is catalyzed by NEPS cyclases and MLPLs, however no enzymes with this function have previously been characterized in other iridoid-producing plants. According to the authors, the ICYCs belong to the methyl esterase (MES) type α/β hydrolases, which are entirely unrelated to Nepeta-specific cyclases, and are phylogenetically well distinguished from other methylesterases. However, they contain functional catalytic triad (serine, aspartate/glutamate and histidine) and a glycine-rich oxyanion hole motifs which are characteristic for canonical esterases, and are responsible for the catalysis of substrate hydrolysis. For the first time, Colinas et al. associate these group of enzymes with the cyclization function, and provide unambiguous proofs for their catalytic activity using in vitro enzymatic assays of the mutant proteins, reconstitution of the secoiridoid pathway in *C. ipecacuanha* and *N. benthamiana*, VIGS silencing of CrICYCs in native plant - *C. roseus*, and the molecular docking analysis. Adopting the split luciferase assays, the authors further prove the interaction of ISY and ICYC, thus suggesting the substrate channeling. The results are well presented and discussed, the methodology is appropriate, and statistical data analysis is adequate. Although the mechanistic basis of the cyclization mediated by ICYCs needs further explanations and experimental proofs, the contents of the manuscript in the present form are original, and will be of interest to a wider community of researchers. The major findings of the Manuscript are novel, and they represent a significant advance in the field, as they fulfil the knowledge gaps about the important steps of the early iridoid biosynthesis pathway in asterids. The results also have applicative potential, as they open a vast array of possibilities for metabolic engineering of plants towards the production of different stereoisomers of iridoids.*

Response: We thank the reviewer very much for the positive feedback and the constructive suggestions to further strengthen the manuscript.

I strongly recommend this manuscript for publication in Nature Plants, pending minor revisions:

-Abstract: The overall impression is that the abstract relies too heavily on previously published literature and does not sufficiently emphasize the novel findings of the current study. I strongly recommend that the authors revise the abstract to clearly highlight their main discoveries, the significance of their results, and the study’s broader contribution to our understanding of iridoid biosynthesis.

Response: We agree with the reviewer that the abstract could be more focused on the presented results. We edited the abstract to change the focus as the reviewer suggests.

In particular, please elaborate on how the knowledge generated in this work could facilitate future efforts to uncover and understand the diversity of iridoids across the plant kingdom, and thus their chemical evolution.

This is a great point: indeed, iridoids are chemically diverse. Although much of this diversity is introduced downstream of the ICYC reaction, our work now explains the dominance of iridoids derived from either of two stereoisomers (7*S*-cis-trans and 7*R*-cis-cis nepetalactol) in plants, a fact that is highlighted in the manuscript. Moreover, it appears that ICYC was lost in the Nepetoideae, which in turn allowed the independent evolution of other iridoid cyclases in *Nepeta*, which in turn lead to increased diversity in nepetalactol stereoisomers. This difference in diversity could not be explained prior to our discovery. These aspects are now included in the main text, though due to the strict word limit, this point is only mentioned in the concluding paragraph. Additionally, because ICYC is the last pathway enzyme before species- or lineage- specific chemically diverse downstream modification it should be an excellent bait gene to find candidates for those downstream genes in a large number of iridoid-producing species.

Be more precise in explaining which reactive intermediates are cyclized by ICYCs, and what is the stereochemistry of resulting iridoids (Lines 33-34).

Response: We agree and have added the names of the reactive intermediate and the specific stereochemistry of the iridoid products to the abstract.

I also suggest to authors to change “iridoid biosynthesis pathway in asterides” into “early iridoid biosynthesis pathway in asterides” (Line 35).

Response: This is an important point, we have changed it to “early biosynthesis pathway...”.

Main: The conclusion paragraph could be improved.

Response: We agree that the conclusion paragraph is somewhat short and have re-written it to further strengthen it while trying to keep the strict word limit of the letter format.

Reviewer #2:

The authors reported the identification and functional characterization of ICYCs, which fill the biosynthetic gap between ISY- and IO-catalyzed reaction steps involved in the biosynthesis of iridoid in asterid plants. The results are interesting for biosynthetic pathway elucidation of plant-derived compounds and the experimentals are well-designed and performed. There are some problems should be clarified.

Response: We thank the reviewer for the valuable feedback and the thorough assessment of the manuscript. We hope that our below answers fully clarify the raised issues.

Major:

1) ISY catalyzes the 1,4-reduction of 8-oxo-generial to form 8-oxo-citronellyl enol. The latter compound is not stable in vitro and a spontaneous cyclization will be taken place, leading to the formation of nepetalactol. However, two kinds of enzymes, MLPL and NEPS, had been reported to catalyze the cyclization of 8-oxo-citronellyl enol to give nepetalactol. NEPS will catalyze the proceeding dehydrogenation of nepetalactol to give nepetalactone as the end product. The authors had clarified that ICYC is not related to NEPS in the manuscript. How about MLPL? Additionally, the novelty of the ICYC-catalyzed reaction is limited since NEPS and MLPL can catalyze the cyclization of 8-oxo-citronellyl enol.

Response: We thank the reviewer for the careful reading of the manuscript and the previously published literature. In the manuscript we mentioned that ICYC is unrelated to the *Nepeta*-specific cyclases (L134-135); specifically, we meant that ICYC (a methylesterase) is unrelated to both NEPS (a short chain dehydrogenase) and MLPL (Major latex protein like).

We agree that this may not be clear and so have now included the names of both classes of *Nepeta* cyclases in the sentence.

In terms of novelty, while, as the reviewer rightly noted, there are previously reported cyclases from the *Nepeta* genus that catalyze cyclization of 8-oxo-citronellyl enol, we would like to emphasize that only ICYC is able to generate 7*R*-*cis-cis* nepetalactol (together with a 7*R*-ISY), which is the precursor for some of the most abundant iridoids in nature (for example catalpol and aucubin). Previous attempts to produce 7*R*-*cis-cis* nepetalactol using various combinations of *Nepeta*-specific cyclases (both MLPL and NEPS) and a 7*R*-ISY failed (see Hernandez-Lozada et al. 2022). Thus, the catalysis of this highly abundant stereoisomer is novel. Additionally, it is entirely unexpected that a member of the methylesterase (MES) family is involved in a non-esterase function. This is highly interesting for researchers interested in specialized metabolism but also important for the broader plant science community as it opens up the possibility that this enzyme family is involved in a range of catalytic transformations. Finally, ICYC is the more common iridoid cyclase in nature, and appears to be present in iridoid producing asterid species across nine orders. In contrast, the *Nepeta*-specific cyclases MLPL and NEPS are restricted to a single genus. In light of all of this, we hope the reviewer agrees that this discovery is impactful and novel.

2) Fig 1e and 3d, there is not any loganic acid was detected by LC-MS from the combination reaction without ICYC. The result is not consistent with that a small amount of nepetalactol produced from the spontaneous cyclization of 8-oxo-citronellyl enol, which will offer a small amount of loganic acid, together with the following enzymes such as IO, 7DLGT, 7DLH.

Response: We thank the reviewer for this careful observation. Previous attempts of iridoid pathway reconstitution in *N. benthamiana* also did not lead to detectable amounts of iridoid glucosides in the absence of a cyclase (see Dudley et al. 2022). Hence, our results are consistent with the literature, which suggests that spontaneous cyclization does not occur at levels sufficient to enable reconstitution of downstream iridoid biosynthesis in a plant host. Even *in vitro*, the level of spontaneous cyclization changes significantly with different buffer conditions (see Lichman et al. 2019). Thus, it is possible that the conditions in planta may simply be unfavorable for spontaneous cyclization.

Minor:

1) Fig 2d, an oxygen atom was omitted in the chemical structure of 7*R*-*cis-cis* nepetalactone.

Response: We thank the reviewer very much for pointing this mistake out. We have corrected the structure in the revised version.

2) L101-102, L128-129, it is not necessary to capitalize the full name of 7DLH and 7DLGT.

Response: We will clarify with the editor what convention should be used here; thank you for pointing this out.

3) L145, the CiIYC should be CiICYC.

Response: We thank the reviewer for noticing this typo and have corrected it.

4) L276-279, the sentence should be revised.

Response: We thank the reviewer for this comment. We have corrected the typo and split the sentence in two sentences for clarity.

5) For the information for references, the authors should check them carefully.

Response: We thank the reviewer for the thoughtful reminder and have double-checked all references and have removed a few that are not relevant.

Reviewer #3:

The manuscript by Colinas and colleagues reports the discovery of iridoid cyclases (ICYC) that catalyze an unusual and hitherto hidden reaction in the biosynthesis of iridoid natural products in asterids. This work illuminates the cyclization of the reactive enol intermediate as a key step in iridoid formation, thus completing the known iridoid biosynthetic pathway. Given the natural functions and medicinal importance of iridoids, this study is timely and should be of keen interest to a broad audience. The presented conclusions are overall well supported, and the experiments have been elegantly performed. The manuscript is very well written and illustrated.

Response: We thank the reviewer for the very positive feedback and for the constructive suggestions to further improve the manuscript.

Below are number of comment that the authors might find helpful to further strengthen the manuscript.

- Line 42: Consider removing 'widespread', since iridoids are largely restricted to asterid species.

Response: We have changed this to "widespread among asterid plants".

- Line 45: Consider adding long-distance vs. local herbivore defenses.

Response: This is a good point. However, we are trying to point out the functions of both iridoid glucosides (the majority of iridoids) as well as the less prevalent volatile iridoids. We think that these encompass both long-distance and local herbivore defenses. While this would be certainly interesting to mention we think that it would require additional sentences to explain these terms to a broader audience. Given the length limitation of the letter format, we find it difficult to accommodate such sentences and have therefore kept the statement fairly general.

- Line 116: Consider specifically mentioning DXS and DXR here, highlighting that the 13 genes include major rate-limiting enzymes.

Response: This is an excellent comment, we have added this to the respective sentence.

- Line 155: Were other metabolic alterations observed that may suggest that accumulating intermediates are metabolized through compatible pathways?

Response: We have not performed a full non-targeted metabolite analysis but our method is able to detect various glycosylated terpenoids. Previously it was shown that silencing of *CrISY* leads to accumulation of various monoterpene glycosides, which was suggested to be a detoxification process of accumulating intermediates including the highly reactive dialdehyde intermediate 8-oxo-geranial (see Geu-Flores et al. 2012). While we detected the previously described peaks when we silenced *CrISY* we did not observe comparable compounds when *CrICYC* was silenced (see base peak chromatogram below, not included in the manuscript).

The accumulating intermediate 8-oxo-citronellyl enol may be less prone to modification by endogenous plant enzymes.

It is also important to emphasize that the VIGS used here is a transient silencing approach that is limited to the first leaf pair that emerges from the plant post-infection. Although genes can be strongly down-regulated, only parts of the tissues show silencing (visible through co-silencing of the marker *MgChel*). Additionally, metabolites can be transported from non-silenced tissues. Therefore, while we can use these experiments to show disruption of production of the expected metabolite, we hesitate to draw conclusions about the effect of this gene silencing on overall metabolism. Despite extensive optimization, we have not been able improve the extent of the VIGS silencing in *C. roseus*, and stable silencing approaches in other Asterid plants have not been reported.

- Line 158: *Supplementary Fig. 10 provides some important insights and might be valuable to be included as a main figure.*

Response: We agree, however, we are unfortunately limited to three main figures for the letter format and have reached the space limit of Fig. 1 where it would make most sense to add. As an alternative to increase visibility of this important result we have converted this SI figure into “Extended Data Figure 2”.

- Fig. 1: *For part (d) it is a bit hard to see how the genes align with the diagram. For part (e) please include a unit for the y axis.*

Response: Regarding d) we did not intend to fully align the gene clusters with the diagram because it makes the synteny look very distorted due to the close proximity of some orders containing orthologs on the phylogenetic tree (for example Cornales and Ericales). To make the figure clearer, we instead color-coded them as the names of the orders. However, we agree that it is not ideal to solely rely on color. Therefore, we have added a fourth column into the diagram called “cluster with *G8H*” which indicates in which orders we found the cluster (indicated by a circle) and for which orders there was no genome data (indicated by asterisk). Additionally, we have converted the grey color of the circles in the diagram to the color coding of the orders. We hope that these adjustments improve clarity of this panel. Regarding panel e) we have added the Y-axis to indicate signal intensity.

- Line 218: *While split luciferase assays should suffice to demonstrate protein interactions in the context of this study, they do not provide evidence for substrate channeling. I would recommend commenting on this more clearly in the discussion. If protein structural and/or subcellular localization data are available, these would provide valuable additional support for this hypothesis.*

Response: This is an important point and we fully agree that an interaction between enzymes does not provide evidence for substrate channeling. We have added a sentence to make this

clearer. The *Catharanthus roseus* ISY ortholog was previously shown to be cytosolic (Geu-Flores et al. 2012) and because we detected interaction between ISY and ICYC, ICYC is likely to be cytosolic as well. Additionally, ICYC is also predicted (DeepLoc) with high confidence to be cytosolic and at least one related alkaloid esterase was shown to be cytosolic (see Colinas et al. 2025). Thus, although we did not perform subcellular localization experiments, it seems likely that both ISY and ICYC are present in the cytosol.

- *Line 261: Is the esterase activity also relevant in planta? For example, did metabolite profiling of VIGS plants indicate alterations in any other than the focal iridoid pathway?*

Response: This is a very interesting point that we could not address fully for reasons explained in response to the point about line 155. We hesitate to draw definitive conclusions regarding this question using a transient silencing method that is limited to a specific tissue. Additionally, the metabolomics method is optimized to detect polar metabolites, and thus we could easily miss effects on more hydrophobic metabolic pathways (e.g. lipids, fatty acids). However, in our limited dataset we did not find any indication of alterations hinting towards esterase activity or other consequences of *ICYC* silencing. We agree that it would be of great interest to conduct more comprehensive metabolomics experiments to detect other possible alterations. However, we believe that such experiments would be most meaningful in stable knockout/knockdown plants which unfortunately cannot (yet) be produced from *C. roseus* or, to the best of our knowledge, any other iridoid producing plant.